

# Biogenic emissions and land-atmosphere interactions as drivers of the diurnal evolution of secondary organic aerosol in the southeastern US

Juhi Nagori[1,2], Ruud H.H. Janssen[3], Juliane L. Fry[4], Maarten Krol[1,2], Jose L. Jimenez[5,6], Weiwei Hu[5,6], and Jordi Vilà-Guerau de Arellano[1]

[1]Meteorology and Air Quality, Wageningen University, Wageningen, The Netherlands
[2]Institute of Marine and Atmospheric Research, University of Utrecht, Utrecht, The Netherlands
[3]Department of Civil and Environmental Engineering, Massachusetts Institute of Technology, Cambridge, MA, USA
[4]Department of Chemistry, Reed College, Portland, OR, USA
[5]Cooperative Institute for Research in Environmental Sciences, University of Colorado, Boulder, CO, USA
[6]Department of Chemistry, University of Colorado, Boulder, CO, USA

*Correspondence to:* Juhi Nagori (j.v.nagori@uu.nl) or Jordi Vilà-Guerau de Arellano (jordi.vila@wur.nl)

**Abstract.**

The interactions between biogenic volatile organic compounds (BVOCs), like isoprene and monoterpenes, and anthropogenic emissions of nitrogen and sulfur oxides lead to high concentrations of secondary organic aerosol (SOA) in the southeastern United States. To improve our understanding of SOA formation, we study the diurnal evolution of SOA in a land-atmosphere

coupling context, based on comprehensive surface and upper air observations from a characteristic day during the 2013 Southern Oxidant and Aerosol Study (SOAS) campaign. We use a mixed layer model (MXLCH-SOA) that is updated with new chemical pathways and an interactive land surface scheme that describes both biogeochemical and biogeophysical couplings between the land surface and the atmospheric boundary layer (ABL).

MXLCH-SOA reproduces observed BVOC and surface heat fluxes, gas-phase chemistry and ABL dynamics well, with the

exception of isoprene and monoterpene mixing ratios measured close to the land surface. This is likely due to the fact that these species do not have uniform profiles throughout the atmospheric surface layer, due to their fast reaction with OH and incomplete mixing near the surface. SOA formation from isoprene through the intermediate species IEPOX and ISOPOOH is in good agreement with the observations, with a mean isoprene SOA yield of 1.8%, and mean monoterpene yield of 10.7%.

However, SOA from monoterpenes, oxidized by OH and $O_3$, dominates the locally produced SOA (69%). Isoprene SOA is

produced primarily through OH oxidation via ISOPOOH and IEPOX (31%).

A sensitivity analysis of the coupled land surface-boundary layer-SOA formation system to changing temperatures reveals that SOA concentrations are buffered under increasing temperatures: a rise in BVOCs emissions is offset by decreases in OH concentrations and the efficiency with which SVOCs partition into the aerosol phase.





# 1 Introduction

Secondary organic aerosols (SOA) produced from the oxidation of volatile organic compounds (VOCs) form an important contribution to aerosol loading (Jimenez et al., 2009; Zhang et al., 2007). They can affect regional climate (Goldstein et al., 2009) and pose health risks to humans (Mauderly and Chow, 2008). A large fraction of SOA is formed by biogenic volatile

organic compounds (BVOCs) which are emitted in large quantities from forested areas, especially during summer (Guenther et al., 1995; Goldstein et al., 2009). Isoprene and the monoterpenes $\alpha$-pinene, $\beta$-pinene and limonene are the most abundant of these BVOCs in the southeastern US (Liao et al., 2007). Consequently, SOA mass in this region has a high biogenic contribution (Ahmadov et al., 2012; Kim et al., 2015).

Anthropogenic emissions can alter the oxidation pathways of biogenic VOCs and thereby the formation of SOA from

biogenic precursors (Spracklen et al., 2011). Recently, the contribution of isoprene to SOA in southeastern US has been studied extensively, with a focus on aqueous-phase reactive uptake mechanisms that are modulated by anthropogenic emissions of sulphur dioxide ($SO_2$) (Hu et al., 2016; Budisulistiorini et al., 2015) . In addition, isoprene SOA can also be produced through condensation of low-volatile organic compounds (LVOC)(Krechmer et al., 2015). Both mechanisms are prevalent under low NO conditions, which are important at the SOAS site, under which the initial oxidation of isoprene by the hydroxyl radical (OH)

leads to the formation of hydroxyhydroperoxides (ISOPOOH) (Paulot et al., 2009), whose oxidation product ($ISOP(OOH)_2$) can condense to form ISOPOOH-SOA. The major channel of ISOPOOH oxidation, however, forms isoprene epoxydiols (IEPOX), which produce IEPOX-SOA upon reactive uptake on acidic surfaces (Krechmer et al., 2015; Hu et al., 2016; Gaston et al., 2014). IEPOX-SOA formation contributed approximately 15-30% to total observed aerosol mass during the SOAS campaign, while ISOPOOH-SOA contributed approximately 2.2% (Lopez-Hilfiker et al., 2016b; Krechmer et al., 2015; Hu et al., 2016).

Monoterpene SOA formation has been shown to be important in the southeastern US (Kim et al., 2015; Zhang et al., 2018) and depends on anthropogenic NOx emissions, which influence daytime oxidation pathways and enhance $NO_3$-radical initiated SOA formation during night-time (Xu et al., 2015; Ayres et al., 2015).

Since SOA concentrations in southeastern US are driven by both natural and anthropogenic factors, understanding future changes in SOA concentrations requires an understanding of these different factors and their interactions. Previous modeling

studies have focused on the effects of future lower anthropogenic emissions of NOx and SOx on the formation of isoprene-derived SOA (Pye et al., 2013; Marais et al., 2016). These studies found that the reductions of anthropogenic emissions of NOx and SOx lead to a net reduction of SOA formation from isoprene.

Here, we study the formation of SOA from biogenic emissions and its diurnal evolution in the context of land-atmosphere coupling, including both biogeochemical interactions (VOC emissions) and biogeophysical interactions (sensible and latent heat

fluxes) between the land surface and the atmosphere, in a case study for the SOAS campaign. The diurnal SOA evolution is driven by ABL dynamics and the interaction between the ABL and the free troposphere (FT), as well as by emissions, chemical transformations and subsequent partitioning into the aerosol phase (Janssen et al., 2012, 2013). Hence, in order to encompass the many aforementioned factors affecting the diurnal SOA evolution, an integrated approach is required to accurately represent the diurnal evolution of SOA and BVOC concentrations.





As sources of SOA, we consider isoprene-SOA formation through aqueous-phase uptake of IEPOX (Marais et al., 2016; Hu et al., 2015) and through condensation of ISOP(OOH)$_2$ (Krechmer et al., 2015), and speciated monoterpene-SOA formation from $\alpha$-pinene, $\beta$-pinene and limonene. We account for the anthropogenic influence on biogenic SOA formation by including the influence of NOx concentrations on peroxy-radical chemistry, in addition to the NOx-induced changing ratios of oxidant concentrations (OH, NO$_3$, O$_3$). We do not include night-time SOA formation.

Our aim is twofold: 1) to improve our understanding of SOA formation in southeastern US from established and recently elucidated pathways, and 2) to understand SOA diurnal evolution in a land-atmosphere coupling context. We build on the case study by Su et al. (2016) that was able to accurately reproduce the dynamics and gas-phase photooxidation of isoprene during the SOAS campaign and then:

1. We couple the dynamics and chemistry of the boundary layer-chemistry model to the land surface and vegetation factors, by including interactive formulations for surface BVOC and heat fluxes.

2. We update the SOA-formation module by including speciated monoterpenes and isoprene-SOA formation through reactive uptake and condensation, to accurately represent the diurnal SOA evolution, as constrained by tower and aircraft observations. Figure 1 shows a schematic of the chemistry mechanism.

3. We study the contribution of different aerosol factors in southeastern US and attempt to identify the source contributions to more oxidised oxygenated organic aerosol (MO-OOA).

4. We analyse the SOA budget and quantify the contribution of different processes and precursors to the SOA diurnal evolution.

5. Finally, we carry out a sensitivity of the integrated land surface-boundary layer-SOA formation system to FT concentrations of SOA and to temperature changes.







**Figure 1.** Formation pathways of secondary organic aerosol (SOA) and interactions included in this study. The atmospheric layers in consideration are shown in blue, and dynamic and surface processes are shown in maroon. The chemical species are in black, the arrows show their movement and the stages the species go through are shown in purple. Biogenic emissions of gas-phase precursors at the land surface are followed by oxidation by OH, $O_3$ and $NO_3$ to form semi-volatile organic compounds (SVOC) which either partition between the gas phase and aerosol phase, condense to the aerosol phase or form aerosol through reactive uptake onto existing acidic aerosol. ISOPOOH and IEPOX are the isoprene oxidation products isoprene hydroxy hydroperoxides (ISOPOOH) and isoprene epoxydiols (IEPOX), respectively. VBS-SOA stands for SOA formed through gas/particle partitioning in the Volatility Basis Set. SOA formation takes place in the Atmospheric Boundary Layer (ABL), which grows in time due to surface fluxes. Entrainment of air from the residual layer brings in aged SOA from previous days and long-range transport.



## 2 Site and Data Description

To constrain and evaluate our model, we use data collected during the Southeastern Oxidant and Aerosol Study (SOAS), held over the period of 1 June to 15 July 2013 (Hidy et al., 2014), and the Southeast Nexus (SENEX) campaign, held in the same time period (Warneke et al., 2016). Both campaigns were part of the Southeast Atmosphere Studies (SAS), which coordinated

comprehensive measurements of trace gas and aerosol compositions, aerosol physics and chemistry and meteorological dynamics across the southeastern US (Carlton et al., 2018). All the measurements (and model) are shown in Central Standard Time (CST).

The case study represents the SOAS main sites near Brent ($32°54'12''N, 87°15'0''W$) and Marion ($32°41'40''N, 87°14'55''W$), Alabama; these are the SOAS ground and flux (above canopy) measurements sites, respectively. A pre-existing South-East Research and Characterization (SEARCH) network site served as the main ground site with gas chromatography-mass spectrometry

(GC-MS) (including speciated monoterpene mixing ratios) (Su et al., 2016) and aerosol mass spectrometry (AMS) measurements (Hu et al., 2015). The National Center for Atmospheric Research C-130 flights collected observations of trace gases, isoprene, monoterpenes, photolysis, methyl vinyl ketone (MVK) and methacrolein (MACR) (Warneke et al., 2016). At the Alabama Aquatic Biodiversity Center (AABC) flux tower (24 km from the Brent ground site and tower) eddy covariance measurements above canopy for surface latent and sensible heat, BVOC fluxes and the shear velocity ($u_*$) measurements were carried out.

We use data from both sides to represent a more regional footprint. Flights with the Whole Air Sample Profiler (WASP) and high resolution proton transfer reaction time-of-flight mass spectrometer (PTR-ToF-MS) measured trace gas concentrations, meteorological data and isoprene and monoterpene mixing ratios above the AABC tower (Su et al., 2016), though speciated monoterpenes mixing ratios are only obtained through GC-MS at the SEARCH site. Data is also used from the NOAA P-3 flights during the Southeast Nexus (SENEX) campaign, which included vertical profile data near the SOAS site (Warneke et al.,

2016). To reduce uncertainties from day-to-day variations and gain representativity, we average the meteorological data and, isoprene and monoterpene emissions and mixing ratios, and trace gas mixing ratios data for 5, 6, 8, 10-13 June following Su et al. (2016). The speciated monoterpene data from the GC-MS measurements are averaged from 5-13 June. WASP research flights were not flown on 7th and 9th June (Su et al., 2016), whereas GC-MS had continuous data.

The total OA concentration measured at the SOAS site by the AMS (DeCarlo et al., 2006; Canagaratna et al., 2007) have

been previously been apportioned by positive matrix factorisation (PMF) to determine the contribution of individual SOA factors (Ulbrich et al., 2009) (See discussion below). The main SOA factors observed at the SEARCH site were isoprene epoxydiol-derived SOA (IEPOX-SOA), isoprene hydroxyhydroperoxide SOA (ISOPOOH-SOA), more oxidised oxygenated OA (MO-OOA), low oxidised oxygenated OA (LO-OOA) and biomass burning OA (BBOA) (Xu et al., 2015; Hu et al., 2015), and the total observed SOA is the sum of all these factors. For this study, the aerosol data is averaged for 6, 8, 10-13 June, since data

is not available for 5th June and incomplete for 7th and 9th June (Xu et al., 2015; Krechmer et al., 2015; Hu et al., 2016, 2015).

## 3 Model Description

We use a mixed layer model for the dynamics of the convective boundary layer with a chemistry and SOA-formation module (MXLCH-SOA) to analyse a representative (diurnal) case study for southeastern US. The model version that we use is described



in Su et al. (2016); Janssen et al. (2013), and a derivation of its basic equations is given in Vilà-Guerau de Arellano et al. (2015). In this section, we summarize the main characteristics of the model and in the following subsections we describe the specific adaptations that have been made for this study, which include new chemical pathways (Section 3.1), SOA formation mechanisms (Section 3.2), interactive BVOC emissions (Section 3.3) and a coupled land surface model (Section 3.4).

MXLCH-SOA approximates ABL mixing under convective conditions (Lilly, 1968; Tennekes, 1973), by assuming vigorous mixing throughout the daytime ABL, resulting in constant mixing ratios with height. The ABL height growth due to entrainment is driven by sensible and latent heat flux (Tennekes, 1973). We consider that the atmospheric boundary layer (ABL) interface with the free troposphere (FT) is an infinitesimal inversion layer with entrainment-driven exchange of scalars and variables between these layers (Tennekes and Driedonks, 1981), i.e. a zero-order closure model. Large scale meteorology is prescribed

based on Su et al. (2016), species segregation is neglected (Ouwersloot et al., 2011) and we do not account for horizontal advection via long-range transport.

    The chemical reaction scheme, which consists of the essential gas-phase reactions of the $O_3$-$NO_x$-VOC-$HO_x$ system (See Table B2), is based on Su et al. (2016) and Janssen et al. (2013). The standard SOA-formation scheme in MXLCH-SOA is based on the volatility basis set (VBS) approach (Donahue et al., 2006).

## 15   3.1   New chemical pathways

We add gas-phase reactions that lead to IEPOX-SOA and ISOPOOH-SOA formation (Reactions 19, 30-34) from Hu et al. (2016). To better represent IEPOX-SOA formation, we included a module for reactive uptake (Section 3.2.2) (Gaston et al., 2014; Hu et al., 2016). The reactions of speciated monoterpenes ($\alpha$-pinene, $\beta$-pinene and limonene) with the three oxidants (OH, $O_3$ and $NO_3$) are also added (Reactions 38-46), as are reactions of isoprene with $O_3$ and $NO_3$ (Reactions 36-37) (Atkinson and Arey,

2003; Orlando and Tyndall, 2012; Crounse et al., 2011; Pye et al., 2010; Wennberg et al., 2018). The $IRO_2$ + $HO_2$ and $IRO_2$ + NO rate constants are updated per Crounse et al. (2011). We use speciated monoterpenes: $\alpha$-pinene, $\beta$-pinene and limonene (the most abundant monoterpenes in southeastern US (Geron et al., 2000)), instead of bulk monoterpene term which is used in Janssen et al. (2012). With these new pathways we can track the actual variability in SOA formation due to different BVOC precursor - oxidant combinations. The contributions to SOA can be quite different depending on the combination, for instance,

limonene + OH or $O_3$ in high $NO_x$ has a yield of 0.62 at $10\mu g$ m$^{-3}$ whereas $\beta$-pinene and $NO_3$ have a yield of 0.26 (Geron et al., 2000; Pye et al., 2010). We also add the BVOC+$NO_3$ oxidation reactions to the VBS module as $NO_3$-initiated oxidation has been shown to contribute substantially to SOA loading (Ayres et al., 2015; Pratt et al., 2012; Fisher et al., 2016). Nitrate radical initiated oxidation is dominant during night-time (as the lifetime of $NO_3$ is very short during daytime), and organonitrate formation peaks at night-time as well (Xu et al., 2015). This is because monoterpene emissions, unlike isoprene emissions,

persist after sundown (Horowitz et al., 2007; Ayres et al., 2015). As the model starts at sun-up and ends in the mid-afternoon, we do not include night-time reactions, though we start with a background SOA that is observed at sun-up.





## 3.2 Secondary organic aerosol formation

In the MXLCH-SOA model we represent the isoprene + OH factors explicitly, using the full mechanism for formation of IEPOX-SOA and ISOPOOH-SOA (Krechmer et al., 2015; Hu et al., 2016), and aggregate the other SOA formation via $O_3$ and $NO_3$ with isoprene and all oxidants with monoterpenes, via the Volatility Basis Set (VBS) partitioning, for comparison with

MO-OOA and LO-OOA (Donahue et al., 2006). However, it is uncertain how much of the aged MO-OOA is locally formed versus advected in via long-range transport. The IEPOX-SOA is formed through reactive uptake and a mechanism to calculate the heterogeneous reaction rate for this formation is included (Gaston et al., 2014; Hu et al., 2016). Lastly, ISOPOOH-SOA formation (upon condensation) is included using reaction rates from (Hu et al., 2016). BBOA is not accounted for in the model, however, as G/P-partitioning depends on the total aerosol mass in the system, it is included in the initialisation of background

SOA. We do not consider isoprene-SOA formed through the methacryloyl peroxynitrate (MPAN) pathway(Kjaergaard et al., 2012), since this pathway had a negligible contribution to SOA formation during the SOAS campaign (Nguyen et al., 2015a), as it is favoured under low temperatures and high $NO_2$ conditions.

### 3.2.1 Gas-particle partitioning

SOA formation through gas-particle (G/P) partitioning in the MXLCH-SOA model follows the Volatility Basis Set (VBS)

approach (Donahue et al., 2006), with semi-volatile products of VOC oxidation lumped into 4 logarithmically spaced bins of effective saturation concentration.

    The SVOC yields for isoprene, $\alpha$-pinene, $\beta$-pinene and limonene are obtained from Pye et al. (2010) and are summarised in Table B6. These yields depend on $NO_x$ concentrations, with the high- and low- $NO_x$ yields interpolated based on the branching reaction of $RO_2$ from isoprene and monoterpene, through NO and $HO_2$ channels. We do not consider G/P-partitioning of the

products of the isoprene + OH reaction since this reaction is explicitly accounted for via ISOPOOH-SOA and IEPOX-SOA formation through condensation and reactive uptake, which are assumed to form low-volatility aerosol products (Hu et al., 2016; Lopez-Hilfiker et al., 2016a). The $\beta$-pinene reaction rates are used here as a proxy for the monoterpene branching and the reaction with NO and $HO_2$ have rates of $k_{\text{TERPRO2NO}} = 2.2 \cdot 10^{-12}$ cm$^3$ molec$^{-1}$ s$^{-1}$ and $k_{\text{TERPRO2HO2}} = 2.1 \cdot 10^{-11}$ cm$^3$ molec$^{-1}$ s$^{-1}$ (Saunders et al., 2003), respectively, while the reaction rates for isoprene are the same, as shown in Table B2. For the enthalpy

of vaporisation we use the recommended value of 42 kJ mol$^{-1}$ from Pye et al. (2010).

    We prescribe a early morning SOA concentration (OA$_{\text{BG}}$) (Janssen et al., 2012), which has the assumed initial value of 3.2 $\mu$g$^{-3}$ in the ABL based on total SOA observations at SOAS (see Figure 7) and 1.5 $\mu$g m$^{-3}$ in FT based on vertical profiles (Figure C4). The effective saturation concentrations are based on Pye et al. (2010) which are more relevant to the southeastern US (Table B6). A deposition velocity of 0.024 m s$^{-1}$ was set for the SVOCs, as per Karl et al. (2010). The dry deposition of SOA

is not considered as it is small, approximately 0.002 m s$^{-1}$ (Farmer et al., 2013).



### 3.2.2 Reactive uptake and condensation

The IEPOX-SOA and ISOPOOH-SOA formation results from the isoprene+OH reaction (R9). The initially formed isoprene peroxy radical $IRO_2$ reacts with OH to give isoprene hydroxyhydroperoxides (ISOPOOH). ISOPOOH reacts with OH and forms either isoprene epoxide (IEPOX) (Hu et al., 2016) or $ISOP(OOH)_2$ (Liu et al., 2016). ISOPOOH-SOA is formed due to

condensation of $ISOP(OOH)_2$ to the aerosol phase, with a yield of 4% (from ISOPOOH + OH). A deposition velocity of 0.03 m s$^{-1}$ is applied for ISOPOOH and IEPOX, as per Nguyen et al. (2015b).

A heterogeneous reaction rate for IEPOX-SOA formation is calculated using a modified resistor model from Hu et al. (2016) and Gaston et al. (2014); a $\gamma_{\mathrm{IEPOX}}$ factor is used to determine the lifetime of IEPOX against aerosol uptake. This factor depends on pH, temperature, particle size, nucleophiles (sulphates and nitrates) and hydrogen sulphate ion ($HSO_4^-$) concentration, and

the mass accommodation coefficient (Gaston et al., 2014; Hu et al., 2016). The IEPOX-SOA was a considerable fraction of the organic aerosol mass measured during SOAS, approximately 17% (Hu et al., 2015), while ISOPOOH-SOA explains a small fraction of aerosol formed through low-NO isoprene oxidation (Krechmer et al., 2015), hence they are included to represent the aerosol composition for the SOAS campaign.

### 3.3 Biogenic volatile organic compound emissions

We implement the Model of Emissions of Gases and Aerosols from Nature (MEGAN) Guenther et al. (2006) to calculate monoterpene and isoprene emission fluxes, driven by light intensity and the temperature of the overlying atmosphere. In this model, the emissions of isoprene and monoterpenes are parameterized depending on base emissions, the production and loss of the BVOC within canopy and the emission activity factors. The base emission rates depend on the plant functional type, which are taken as a broad leaf forest at the SOAS site (Guenther et al., 2006). The isoprene fluxes are light dependent so we use

the parametrized canopy environment emission activity (PCEEA), and we use air temperature instead of skin temperature in our formalism, as the PCEEA already accounts for the canopy temperature being higher than air temperature (Alex Guenther, personal communication 2017). The daily average photosynthetic photon flux density (PPFD) was calculated between 400 to 500 $\mu$mol m$^{-2}$ s$^{-1}$ for this site (Alex Guenther, personal communication 2017). We use a conversion factor of 4.766 to convert the photosynthetically active radiation (PAR) value from W m$^{-2}$ to PPFD above canopy in $\mu$mol m$^{-2}$ s$^{-1}$, per the Goddard Earth

Observing System chemistry (GEOS-chem) model. We calculate the monoterpene flux depending on the canopy emission activity factor and the soil moisture emission activity factor and use skin temperature instead of air temperature (Guenther et al., 1995). Table B5 summarizes the MEGAN parameters applied here.

To derive speciated monoterpene emissions, factors of 45%:45%:10% are applied to allocate the emissions to $\alpha$-pinene, $\beta$-pinene and limonene, respectively, based on their average relative abundances observed during the SOAS campaign as per

figure S1 in Ayres et al. (2015).



### 3.4 Coupled Land-surface model

A land surface model (Van Heerwaarden et al., 2009) is coupled to MXLCH-SOA to enable the interactive calculation of surface heat fluxes, based on the Penman-Monteith equations for evapotranspiration (Monteith et al., 1965). With this inclusion, MXLCH-SOA can be used to simultaneously and interactively calculate the exchange of energy (sensible heat flux) and water

(latent heat flux) between the land surface and the ABL. These heat fluxes, in turn, drive the diurnal dynamics of the ABL.

In this way, an online coupled land surface-ABL-SOA formation model is obtained, in which the exchanges of energy and VOCs between the land surface and the ABL at the diurnal time scale are internal variables of the coupled system. This means that only forcings (drivers external to the system at the appropriate time scales) are prescribed to the model. Note that dry deposition is not yet calculated interactively, we instead utilise deposition velocities from other literature. We evaluate the

interactively calculated surface moisture and heat fluxes with the eddy covariance measurements taken at the AABC tower.

Table B4 shows the land surface characteristics used to calculate the dynamic fluxes interactively, where typical values for broadleaf trees are used. We model above canopy and include a wind module where the initial U-wind and V-wind are set at 1 m/s. These wind module values are used so as to have a more realistic value of the aerodynamic resistance, $r_a$, which is otherwise very large in the first time step due to a very small convective velocity scale, $w_*$. The $r_a$ is inversely proportional to

$w_*$ in the model.

### 4   Numerical experiments

We use the MXLCH-SOA model to perform a set of numerical experiments to improve our understanding of SOA formation during SOAS in a land-atmosphere coupling context. First, we set up a base case, by expanding the case study of Su et al. (2016), guided by the observations of heat and VOC fluxes, ABL dynamics, and VOC and SOA concentrations. We then evaluate the

contributions of the different dynamical and chemical processes to the diurnal evolution of the SOA concentration, and dissect the SOA budget to show the contributions of the various precursors and chemical pathways to SOA formation.

The dynamical initial and boundary conditions for the base case are shown in Table B1, and are based on Su et al. (2016). We apply a lapse rate of 0.002 K m$^{-1}$ below 1150 m and 0.005 K m$^{-1}$ above 1150 m to better constrain the boundary layer height (See Figure 3). The lapse rate mimics upper air conditions and counteracts the development of the ABL, and we adjust this

value so that the observed evolution of the boundary layer was satisfactorily reproduced by the model (See Figure 3). The initial conditions for the chemical species are based on the observations from the SEARCH and AABC sites and Su et al. (2016) (See Table B3). The initial concentrations of SOA in the boundary layer are based on the AMS observations taken at the SEARCH site (Hu et al., 2015).

After establishing the base case, we carry out a series of numerical experiments to assess the impact of FT concentration of

SOA on the diurnal SOA evolution to stress the importance of information of early morning residual layer concentrations. We use FT concentrations, as we have a few measurements of SOA concentration above the boundary layer at 11:00 CST from SENEX flights (Warneke et al., 2016). We use the range of these observations as constraints for the numerical experiments.



Finally, we explore the effect of a changing climate on the near-surface SOA concentration. Our main interest is in improving our understanding of the net effect on SOA concentrations of several interacting processes that can either reinforce or compensate for each other. The increase in average air temperature under a warmer climate has several effects on the coupled system that may affect SOA concentrations: 1. VOC emissions increase, 2. the partitioning efficiency of SVOCs into the aerosol phase

decreases and 3. the vapour pressure deficit (VPD) decreases, which modulates the heat fluxes and consequently the boundary layer height (Van Heerwaarden et al., 2009). We simulate a warming climate of 1 and 2 K. For this purpose, the early morning values of mixed-layer temperature, surface temperature, and soil temperature in both layers are all increased (and decreased) by 1 and 2 K. In order to stay consistent with climate warming predictions, the initial relative humidity is kept constant, which is done by calculating the values of the specific moisture at each temperature increment using the Clausius Clapeyron relation

((Van Heerwaarden et al., 2009)). In this way the sensible heat flux forcing is more consistent with future climate warming. As previous literature ((Hansen et al., 1999, 2001; Goldstein et al., 2009)) has observed the southeastern US to have undergone a cooling trend compared to the rest of the US in the summer months, we add 2 more runs with a cooling of 1 and 2 K, respectively.

## 5   Results

### 5.1   Surface heat and BVOC Fluxes

We are able to successfully represent the dynamics, surface conditions and gas-phase chemistry and hence have a good balance of the three in this model. The correspondence of the model to those observations is comparable to Su et al. (2016).

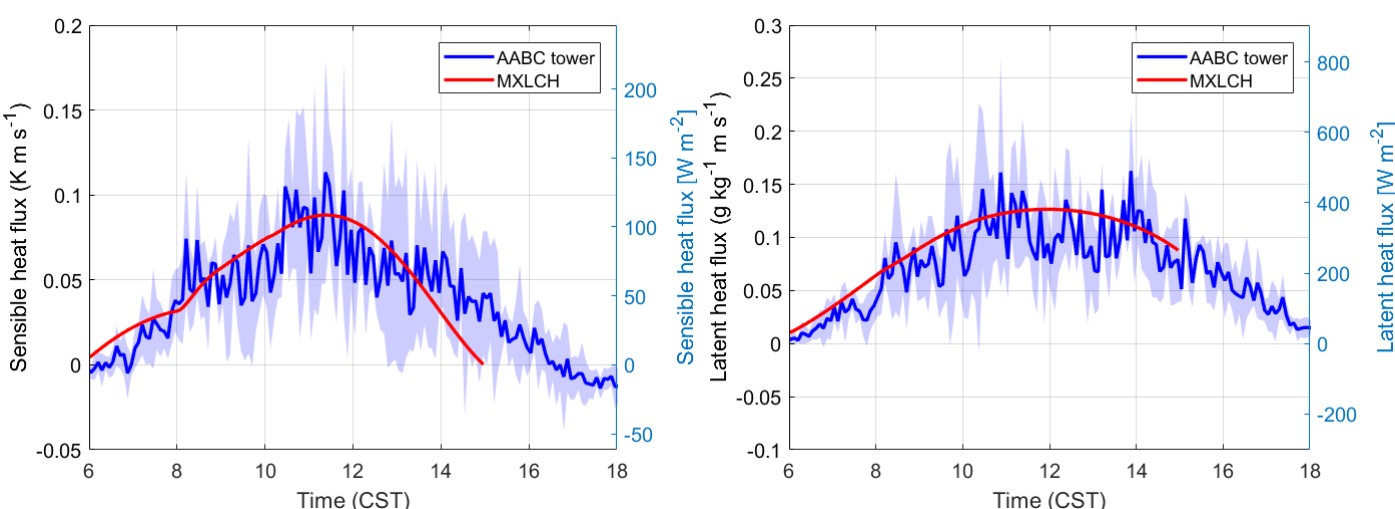

**Figure 2.** (a) Sensible and (b) latent heat flux measured (blue) and modelled (red) at the Alabama Aquatic Biodiversity Centre (AABC) eddy-covariance tower. The blue shaded area represents the data variability over over June 5, 6.8, 10-13, 2013, while the solid blue shows the average over these days.





Figure 2 shows that the interactively calculated sensible and latent heat fluxes match well with the observations. The modelled sensible heat flux peaks before noon (at around 100 W m$^{-2}$), and is underestimated compared to the observations at the end of the afternoon. However, measurements are largely in the range of observations and eddy-covariance measurements have an uncertainty range of approximately 15-20%, as measurements mostly underestimate the fluxes (possibly due to unresolved

eddies) (Field et al., 1992; Weaver, 1990). The modelled latent heat flux matches the observations, and peaks at noon (just below 0.14 g kg$^{-1}$ m s$^{-1}$ or 400 W m$^{-2}$). The Bowen ratio (the ratio of the sensible heat to the latent heat) is consistent with being above a moist surface, as the latent heat flux is larger than the sensible heat flux.

The dynamics are also successfully represented; the boundary layer height is well within the range of observations (Figure 3). The boundary layer is shallow in the early morning and its height increases rapidly between 08:00 and 10:00 CST from 400 m

to about 1100 m, after which it slowly rises to 1300 m by 14:00 CST. The rapid increase between 08:30 and 10:00, once the capping inversion is overcome, is due to the peak in the entrainment flux which adds heat and dry air to the boundary layer from the RL, resulting in the rapid growth of the boundary layer (Vilà-Guerau de Arellano et al., 2009).

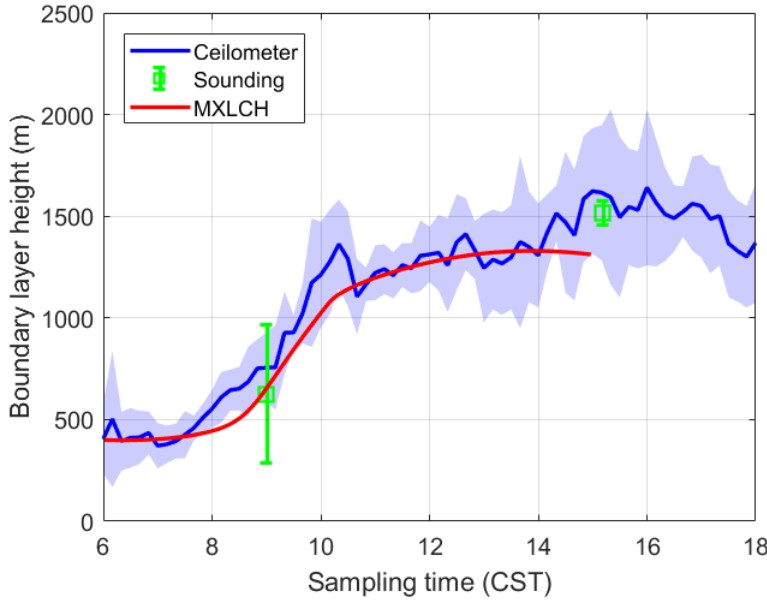

**Figure 3.** Boundary layer height measured (blue and green) versus modelled by MXLCH model (red) over the SOAS super site during the SOAS measurement campaign for the days 5, 6, 8, 10-13 June, 2013.

Figure 4 shows the interactively calculated above-canopy monoterpene and isoprene emissions (calculated from Appendix B). The isoprene flux falls in the lower end of the measurements (but within their uncertainties), while the monoterpene emissions

are modelled accurately compared to the observations. The isoprene flux peaks at noon (at 1.1 ppb m s$^{-1}$) while the monoterpene emission flux peaks at noon at just 0.05 ppb m s$^{-1}$. The diurnal range of monoterpene emissions is small compared to isoprene (only 0.03 ppb m s$^{-1}$), because monoterpene emissions depend only weakly on light (Emmerson et al., 2017). On the other hand,





isoprene emissions respond to the diurnal light availability (Guenther et al., 2006). Hence, the emission rates for isoprene are much more variable than emission rates for monoterpenes, as the model is run during the day, with abundant light availability (measurements are chosen from clear days). Monoterpene emissions are mainly temperature dependent, hence there is a slight increase towards noon (Holzinger et al., 2005). As outlined before, we speciate the monoterpene emissions as 45% $\alpha$-pinene, 45% $\beta$-pinene and 10% limonene.

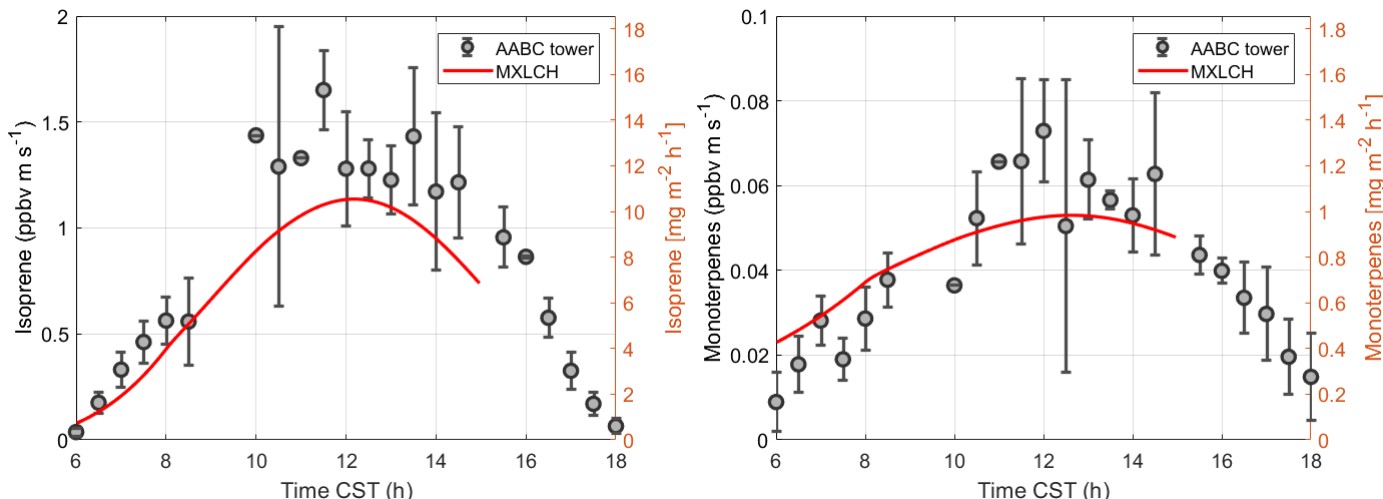

**Figure 4.** Measured (black) and modelled (red) (a) isoprene and (b) monoterpene fluxes at the Alabama Aquatic Biodiversity Centre (AABC) eddy-covariance tower for June 5, 6.8, 10-13 June, 2013.

## 5.2 Diurnal evolution of BVOC mixing ratios

Figure 5a shows the mixing ratio of the bulk monoterpenes (sum of the mixing ratios of $\alpha$-pinene, $\beta$-pinene and limonene). The initial value is 1.0 ppb which decays rapidly until 10:00, followed by an increase to just above 0.25 ppb at the end of the day. This shape of the monoterpene is reflected in the respective shapes of $\alpha$-pinene, $\beta$-pinene and limonene (See Figure C2). The decay rate of the modelled monoterpenes is much higher than the surface observations (blue - SEARCH tower). The model underestimates the monoterpene mixing ratio compared to these ground observations (by about 0.5 ppb; almost by a third). These GC-MS measurements are taken on top of the SOAS tower, which is just above canopy height (20 m). The difference in the model and measurements might arise since the measurements are done within the roughness sub-layer, which is 3 times the canopy height ($h_c$) (Vilà-Guerau de Arellano et al., 2015). The MXLCH model assumes a well-mixed ABL, with a coupled surface layer model. However, concentrations closer to the surface fall within the roughness sub-layer are usually different than in the mixed layer (Stull, 1988).





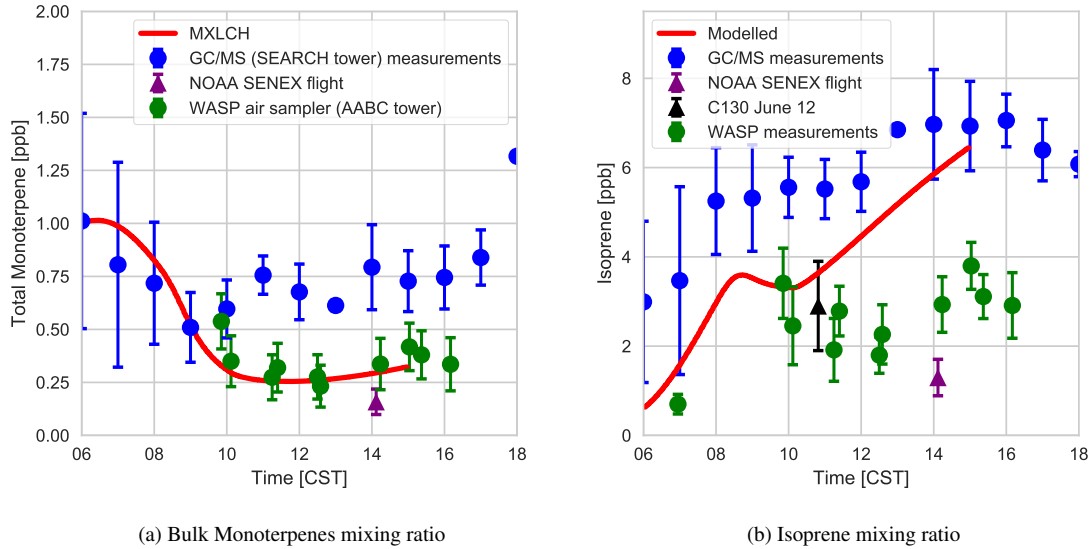

| (a) Bulk Monoterpenes mixing ratio | (b) Isoprene mixing ratio |

**Figure 5.** (a) Total monoterpenes (sum of $\alpha$-pinene, $\beta$-pinene and limonene) and (b) isoprene measured above canopy (blue) above the SEARCH tower averaged from 5-13 June 2013, vertical profile measurements made by the NOAA SENEX campaign flight (purple) averaged for 11 June above the SEARCH super site, by whole air sample profilers (WASP) - (green and averaged for 5, 6, 8, 10-13 June) and by the NCAR C130 flight (black) on 12 June 2013 above SEARCH super site. MXLCH model output in red. Error bars indicate one standard deviation. The WASP sampler only measured the bulk monoterpene mixing ratio instead of the speciated monoterpenes.

The mean values calculated from the vertical profiles of monoterpenes (made by SENEX above SEARCH; black and purple Figure 5a and WASP above the AABC tower - green) indicate lower mixing ratios compared to surface (GC-MS) measurements (blue Figure 5a). These vertical profiles agree much better with the mixed layer approximation, with very good representation of model with WASP air sampler measurements. As we are modelling the air above the canopy, airplane measurements give a good

5    average of measurements in the atmospheric boundary layer, leading to better representativeness. The WASP air sampler only measures the bulk monoterpenes and not for speciated monoterpenes, so a comparison per monoterpene cannot be made as in Figure C2.

The isoprene mixing ratio (made by WASP and NCAR-130 above AABC; green and black Figure 5b) match well with the vertical profiles at the start of the day. However, they are overestimated in the late afternoon (boosted due to the high emissions

10    calculated above the AABC tower; modelled 6.4 ppb while measurements indicate 4 ppb $\pm 0.8$ ppb) and match the surface measurements (Figure 5b) taken at the SEARCH tower better (blue). The difference in measured isoprene mixing ratios indicates that isoprene is not very homogeneously well mixed in the horizontal or vertical.

According to Su et al. (2016), ground based measurements of species with short lifetimes (as is the case for monoterpenes and isoprenes) are not representative of the averaged concentrations inside the CBL. A short chemical lifetime could explain





the disparity between the mixing ratio of the monoterpenes and isoprenes at the surface and measured in the vertical profile. According to Holzinger et al. (2005), the monoterpene concentration peaks in less well-mixed conditions (especially at night) and in more well-mixed conditions the monoterpene concentration falls. The oxidative lifetime of monoterpenes is relatively short; monoterpene lifetime is between 18-48 minutes (Holzinger et al., 2005). Isoprene has a lifetime of approximately 1.4

5  hours (Xu et al., 2015). However, these times are comparable with the turbulent mixing time scales which is calculated as the boundary layer height divided by the convective velocity scale ($w_*$). This velocity scale depends on the buoyancy of the air parcel and determines the time taken for the air parcel to reach the boundary layer (Vilà-Guerau de Arellano et al., 2015), which in this model is between 20-40 minutes, which is comparable to the monoterpene lifetime.

In summary, within the limits of the measurements and observations, we obtained a reasonable representation of the diurnal

10  evolution of the gas-phase composition in a dynamically evolving boundary layer. Moreover, the evolution of other gas-phase mixing ratios are also reproduced within measurement range (Figure C1). Next, we investigate the SOA concentration and diurnal evolution.



## 5.3 Diurnal evolution of Isoprene SOA

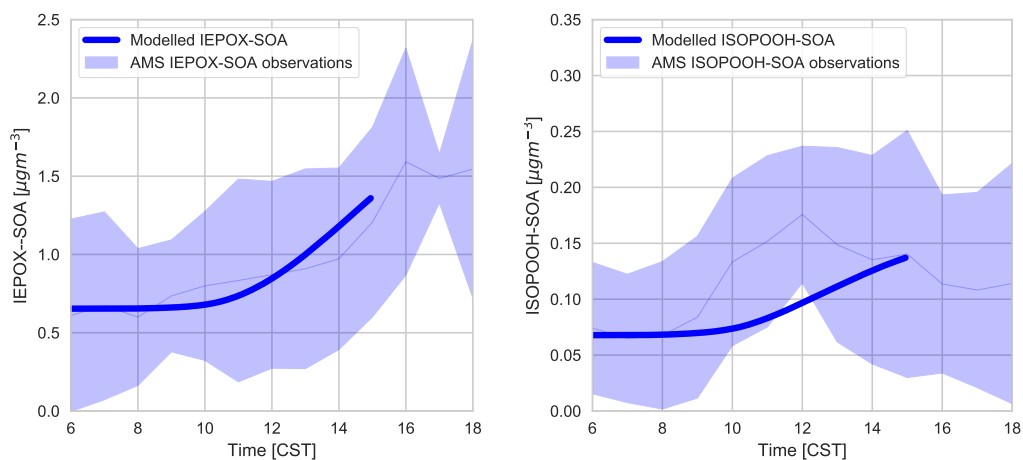

**Figure 6.** (a) Diurnal IEPOX-SOA evolution measured with aerosol mass spectrometry versus modelled IEPOX-SOA and (b) diurnal ISOPOOH-SOA evolution measured with aerosol mass spectrometry versus modelled ISOPOOH-SOA above the SEARCH super site and the light blue shaded area represents the variability over these days (measurements averaged over 6, 8, 10-13 June).

Figure 6 shows that the model is able to capture the observed evolution of both IEPOX-SOA and ISOPOOH-SOA, which is similar to Hu et al. (2016) and Krechmer et al. (2015), respectively. The concentrations of IEPOX-SOA and ISOPOOH-SOA increase throughout the day, following the isoprene mixing ratio (Figure 5b). At the end of the day the IEPOX-SOA

concentration is 1.45 $\mu$g m$^{-3}$, while ISOPOOH-SOA concentration equals 0.155 $\mu$g m$^{-3}$. There is a peak at noon in ISOPOOH-SOA measurements which matches with Krechmer et al. (2015), but this peak is not captured by the model. ISOPOOH-SOA formation depends on the OH concentration and hence the fast rise in ISOPOOH-SOA coincides with the OH peak. The ISOPOOH-SOA is otherwise within the range of observations. IEPOX-SOA formation is faster after noon, due to a peak in OH concentration and isoprene emissions. From the ISOPOOH formed from this reaction, the branching ratio to IEPOX and

ISOP(OOH)$_2$ is approximately 88% and 2.5%, which results in larger concentration of IEPOX-SOA compared to ISOPOOH-SOA (Krechmer et al., 2015). The mean isoprene SOA yield (the amount of IEPOX-SOA and ISOPOOH-SOA produced compared to the total isoprene chemical loss in the model) was calculated at 1.8%, which is lower compared to the 3.3% calculated by Marais et al. (2016) but well within the range of 1-6% discussed by Krechmer et al. (2015).

The calculated $\gamma_{\text{IEPOX}}$ in ambient SOAS conditions was 0.0087, and the subsequent heterogeneous reaction rate was calculated

at $1.5 \cdot 10^{-4}$s$^{-1}$ and agreed with Hu et al. (2016). This value successfully models the observed IEPOX-SOA. The IEPOX lifetime to uptake on acidic aerosol is relatively slow (timescale of approximately 5 hours), though it depends on the time of the day. pH is low in the afternoon and this accelerates uptake (Krechmer et al., 2015). We use a pH of 0.8 (corresponding to Hu et al. (2016)



where the H⁺ proton concentration = 0.15 M for the ambient case), though as we do not include diurnal variation of pH in this model, the diurnal effect is not captured in the model. The relatively slow uptake implies that dry deposition and OH reaction compete significantly with the heterogeneous uptake of IEPOX, as concluded in prior studies (Hu et al., 2016; Nguyen et al., 2015a). The budget contribution of IEPOX-SOA to total SOA is small in the first three hours of the day, and picks up in the

latter part of the day, which follows the isoprene peak. The rate of IEPOX-SOA formation peaks at 14:00 CST, with the steepest increase between 10:00 and 12:00 CST. Once formed, IEPOX-SOA is thought to have a relatively long lifetime (1-2 weeks against wet deposition, 2 weeks through heterogeneous OH reaction (Hu et al., 2016)).

## 5.4  Constraining the SOA budget at SOAS: model versus observations

Figure 7 shows the diurnal evolution of the measured, total SOA (and the contribution of each observed factor) against the

modelled IEPOX-SOA, ISOPOOH-SOA and the modelled total SOA (as a sum of IEPOX-SOA, ISOPOOH-SOA and MT-SOA). The light blue shaded area shows the variability over the days averaged of the aerosol measurements, and the modelled diurnal evolution of the SOA falls within this standard deviation. The modelled SOA concentration remains relatively constant in the early morning, as is reflected by the SOA observations. The modelled SOA concentration then decreases as it is diluted by the ABL growth as entrainment mixes in air with a lower SOA concentration. The modelled SOA concentration increases towards

the end of the day, driven by the rise in ISOPOOH- and IEPOX-SOA concentrations, reaching 3.5 $\mu$g m$^{-3}$ by the end of the day.

According to the model, the largest contribution to SOA comes from gas-aerosol partitioning of monoterpene oxidation products, approximately between 73% in the morning and 58% by the end of the day with a mean of 69%. This monoterpene SOA can be compared to the LO-OOA and the MO-OOA measurements, though MO-OOA is assumed to be more aged and could either be left over from the previous days (entrained from the RL), a result of advection, in which case it is not locally

produced and represents a regional concentration (Xu et al., 2015; Jimenez et al., 2009), or a result of fast oxidation (and hence locally produced). Monoterpene-SOA is formed via gas-particle partitioning and Figure C3 shows the partitioning that takes place in each of the four bins in the VBS.

Based on the PMF source apportionment, the LO-OOA and MO-OOA contributed respectively 33% and 39% to ambient total SOA in southeastern US (Xu et al., 2015). Hence, throughout the campaign a major part of SOA is LO-OOA and MO-OOA in

southeastern US and hence a large part of SOA formed in the model can be attributed to G/P partitioning. As the majority of the G/P partitioning monoterpene based, monoterpene derived SOA contributes significantly to total SOA formation in southeastern US (Ayres et al., 2015; Zhang et al., 2018). Addition of nitrate reactions can also have a significant contribution to the SOA fraction (Ayres et al., 2015), however, this is not the case in our model, as observed in Figure C6.

The model, which predicts locally formed OA only, bisects the MO-OOA between 09:00 and 15:00 (Figure 7), implying that

there is some aged SOA in the system (more than 1 $\mu$g m$^{-3}$ in the system at 11:00). In the morning, as there is not much OH history, the aged MO-OOA could be from the previous day, and entrained into the ABL from the FT. In the afternoon, the ABL stops growing, and is deeper, such that local effects become more dominant. Local partitioning of SVOC contributes between half and the majority of the MO-OOA in the afternoon, which could indicate that aerosol becomes more aged over the day, in approximately 4 hours. It would be instructive to study changes in the composition of the species comprising MO-OOA with



more molecularly-specific analysis methods, and check whether this change over the day is consistent with a shift from aged to rapidly-oxidised local product or whether it is just an identical product mixture from a different region. This might answer the question of aged SOA transported in versus fast, local oxidation. Most importantly, however, the model and measurements agree on 3-4 $\mu$g m$^{-3}$ of afternoon SOA during the SOAS campaign.

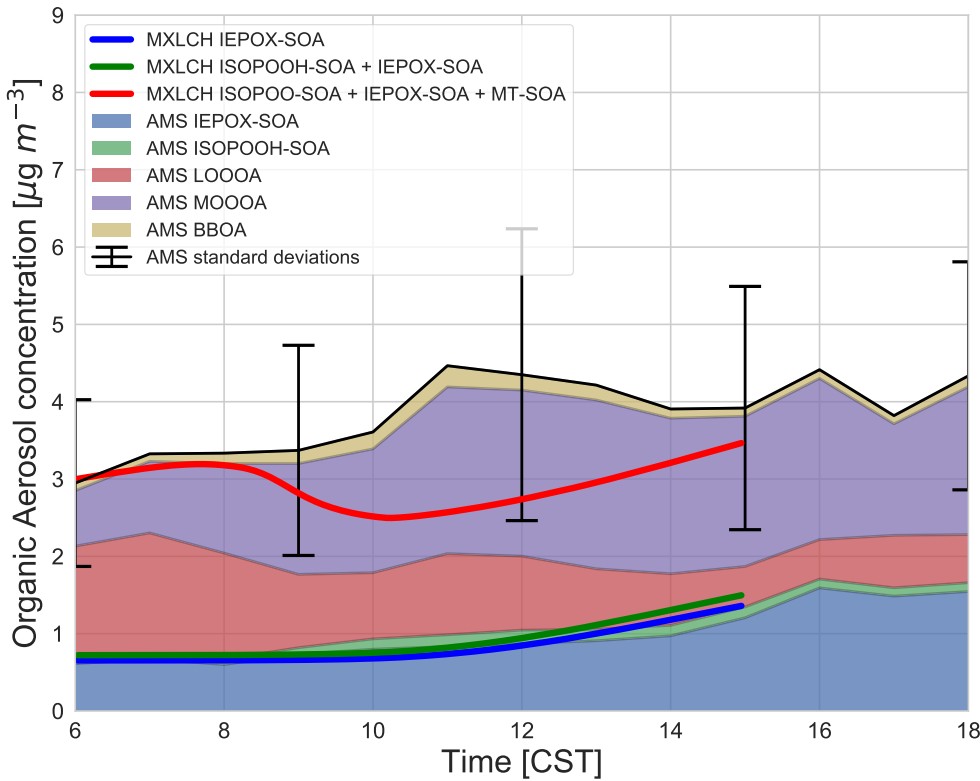

**Figure 7.** SOA measured at the SOAS site versus SOA modelled in the MXLCH model. The observations are averaged over 6, 8, 10 - 13 June 2013 and show the stacked contribution of IEPOX-SOA, ISOPOOH-SOA, LO-OOA, MO-OOA, and BBOA, which made up the majority of the aerosol mass at SOAS site. The light blue area shows 1 standard deviation of the total SOA measurements. The solid lines show the SOA modelled in MXLCH-SOA, with the blue line showing IEPOX-SOA, the green shows IEPOX-SOA + ISOPOOH-SOA and the red line shows the total SOA (IEPOX-SOA + ISOPOOH-SOA + MT-SOA) formed in the model.





## 6   Budget analysis

A bulk budget analysis can be used to differentiate the contribution of entrainment and the different SOA factors to the SOA budget.

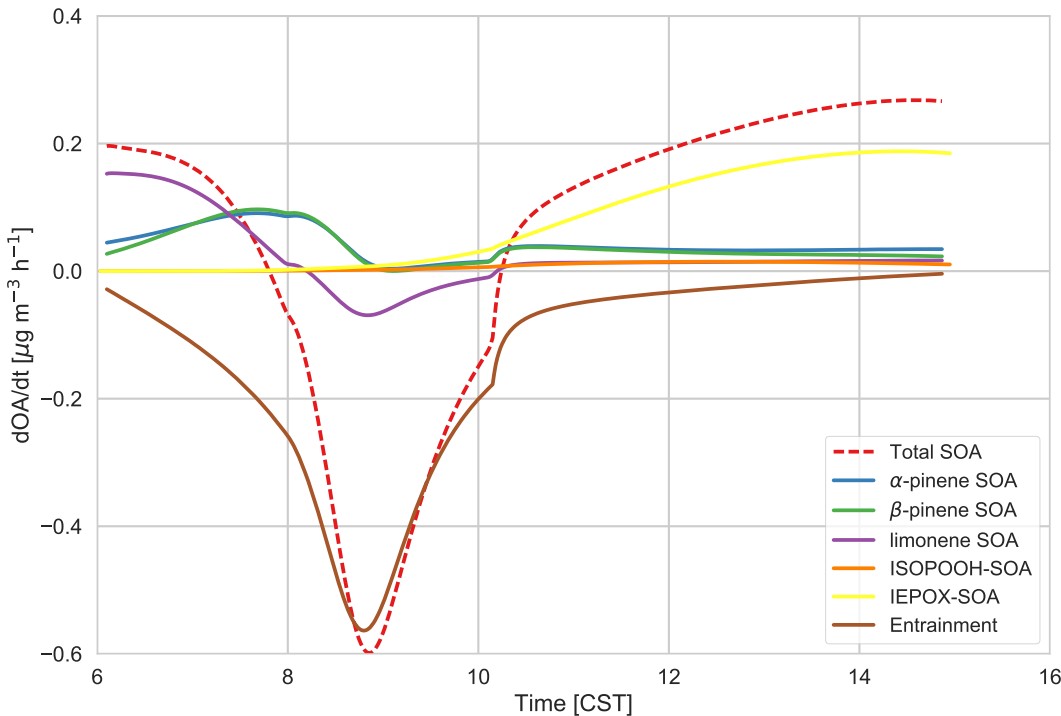

**Figure 8.** The SOA budget, which consists of of the total tendency (dashed), contribution from entrainment of background OA (pink), the chemistry contribution is split into IEPOX-SOA (yellow), ISOPOOH-SOA (orange), and $\alpha$-pinene SOA (red), $\beta$-pinene SOA (blue), limonene SOA (green). The total MT-SOA (purple) is just the sum of $\alpha$-pinene SOA (red), $\beta$-pinene SOA (blue), limonene SOA (green).

From Figure 8, we can determine the contributions of different processes and chemical species to total SOA. The early morning SOA consists primarily of MT-SOA (formed by gas/particle partitioning) as per Figure 7. The contribution of entrainment to the total rate of change of the SOA concentration peaks at 09:00, where entrainment contributes more than 86% to the total SOA tendency. Hence, as the boundary layer height is growing the fastest (Figure 3) and the entrainment velocity is also peaking (0.12 m s$^{-1}$ at 09:30), the SOA concentration decreases due to the introduction of SOA-poor air from the RL. Just after 10:00 CST, the effect of entrainment is low and hence the SOA tendency becomes positive again, as production picks up from a sum of IEPOX-SOA, ISOPOOH-SOA and MT-SOA from G/P partitioning. By the late afternoon, IEPOX-SOA has the largest contribution to the SOA budget (68%), while the contribution of MT-SOA decreases (to 27%) at this time, as the





contributions of $\alpha$-pinene, $\beta$-pinene and limonene are lower in the afternoon. The mixing ratios of these monoterpenes are also lower in the afternoon as not much is added as most is reacted away in the morning, hence the contribution to SOA is smaller. ISOPOOH-SOA has a a very small contribution to the SOA budget (end of day contribution 3.9%).

$\alpha$-pinene contributes about 18% of the total SOA, while $\beta$-pinene contributes about 10% in the early morning. The contribution of both rises and by 08:00, the MT-SOA is largely from $\alpha$- and $\beta$- pinene ($\alpha$-pinene and $\beta$-pinene are approximately 50% each). By the end of the day, $\alpha$-pinene SOA dominates, contributing about 50% to the MT-SOA and 12.5% to the total SOA. Rather surprisingly, the limonene product dominates the SOA contribution in the morning (approximately 60%). This is surprising as the limonene mixing ratio is much lower than the $\alpha$- and $\beta$- pinene (Figure C2). However, as discussed in previous literature (Lee et al., 2006; Krechmer et al., 2015), limonene-SOA yield is much higher than the yield of $\alpha$-pinene and $\beta$-pinene. The stoichiometric coefficients for limonene + OH and $O_3$ are also higher than OH and $O_3$ + $\alpha$- and $\beta$- pinene stoichiometric coefficients. As there is an OH peak in the morning in the shallow boundary layer, and the oxidation reactions between limonene and OH and $O_3$ are fast (Table B3: R44, R45) this results in a large accumulation of limonene SOA product in the morning. As the boundary layer grows, entrainment dilutes this product causing a fall in the limonene SOA tendency. As the day progresses the contribution of limonene SOA becomes less dominant (6% by the end of the day) and the $\alpha$- and $\beta$- pinene contributions become more important. The isoprene + $O_3$ and $NO_3$ pathways lead to a negligible amount of SOA formed in our model. The oxidant + BVOC pathway contribution can be seen in Figure C6; OH-oxidation is the most important contributor to aerosol formation.

## 7 Sensitivity analysis: early morning SOA profile

To test the sensitivity of the coupled land surface-boundary layer-SOA-formation system, we carried out numerical experiments on initial conditions of the model. We evaluated the effect of the initial RL concentration of SOA on the diurnal evolution of SOA in the ABL.

These experiments are guided by measurements of SOA concentration above the boundary layer at 11:00 CST from the SENEX flights (Figure C4). We use the range of these profile measurements as constraints on the numerical experiments. In the previous section, we discussed the entrainment of aged SOA from previous days from the RL into the mixed layer as the boundary layer grows. Figure 9 shows the sensitivity of the diurnal SOA evolution in the boundary layer to the concentration of background SOA in the RL. We constrain SOA concentrations by the vertical profiles from by the SENEX flights (Figure C4 and Wagner et al. (2015)) and a case where the concentration of SOA is the same in the ABL and RL at the start of the simulation. We compare the effect of the RL SOA concentration on the modelled SOA against the observed SOA concentrations.

We find that a uniform SOA concentration in the ABL and RL no longer leads to a drop in SOA due to growth of the ABL, but also leads to overestimates compared to the observations during the end of the afternoon. In cases where the concentration of SOA is less in the RL than the ABL, there is a dilution of SOA as the boundary layer grows, as entrainment mixes air with less SOA from the RL. This is more marked when the concentration difference is larger. This difference is also found by Janssen et al. (2012, 2017), who discussed the importance of background OA concentration in the RL; if there is a large jump of background



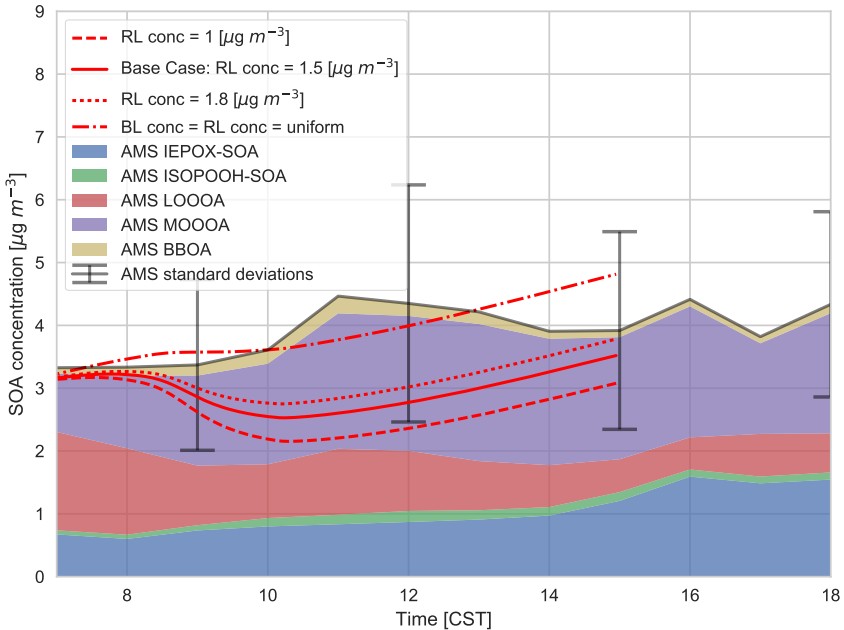

**Figure 9.** Sensitivity of the diurnal SOA evolution to initial free tropospheric organic aerosol (OA$_{BG}$), compared to the (average) AMS observations of IEPOX-SOA, ISOPOOH-SOA, LO-OOA, MO-OOA and BBOA, averaged over 5, 6, 8, 10-13 June, 2013. The BL and RL conc indicate the concentrations in the boundary layer and residual layer, respectively.

OA between the ABL and RL it has a significant effect on diurnal SOA evolution. Tracer concentrations are generally lower in the RL compared to the ABL (which is the case for SOA in Figure C4), and hence entrainment dilutes the concentrations in the ABL (Karl et al., 2007, 2009). Hence, in order to accurately understand the diurnal SOA evolution, it is very important to have a good estimate of its RL concentration in the early morning.

5    This sensitivity analysis also gives an opportunity to allocate the source of SOA. As SOA is relatively long-lived, the amount of aged SOA in the RL can have a large effect on the SOA in the ABL, as it affects the vertical mixing of SOA and SOA availability for G/P partitioning. If we consider a uniform concentration in the ABL and RL, it seems like most of the MO-OOA is captured by the model (implying dominance of local production), though there is an overestimation of SOA formation in the early morning and late afternoon (although the model results are within one standard deviation of the measurements and within measurement uncertainties). Since the available measurements show that the OA concentration in the RL is between 1-1.8

10    $\mu$g m$^{-3}$ (Figure C4 and Wagner et al. (2015)), not all the aged MO-OOA can be explained by this process and some must be horizontally advected. In addition rapid formation of MO-OOA via autoxidation reactions (Ehn et al., 2014), or the substantially lower volatility of ambient SOA compared to that assumed in the VBS-based models (Hu et al., 2016; Stark et al., 2017) may





contribute to explain the model-measurement differences in MO-OOA when the experimentally-constrained FT concentrations are used in the model.

# 8 Sensitivity analysis: SOA formation in a changing climate



**Figure 10.** The response of isoprene emissions (top-left), gas-phase IEPOX mixing ratios (top-middle), OH (top-right), total SVOC mixing ratios (bottom-left), partitioning efficiency in the first volatility bin (bottom-middle) and the total SOA concentration (bottom-left) to changing temperatures.

Using our coupled land surface-boundary layer-SOA formation model, we can study the net effect that temperature has on
5   SOA concentration through VOC-emissions, G/P-partitioning, and through feedbacks between the ABL and the land surface that influence entrainment of SOA from the residual layer. In our experiments, in which we varied the early morning temperature





by between -2 and +2 K, we find that the total SOA concentration in the daytime ABL is buffered against temperature changes (Figure 10).

Isoprene (Figure 10 top-left) and monoterpene emissions (Figure C5 bottom-left) are temperature dependent (Guenther et al., 2006), and consequently we observe positive impact of rising temperature on these BVOC fluxes. At higher temperatures, this

means there is an accumulation of BVOCs in the ABL, which consequently leads to a depletion of OH (Figure 10). However, as we do not take OH-recycling into account in the oxidation of isoprene, this has an effect on the OH depletion. The change in the IEPOX gas-phase mixing ratio (Figure 10) is not as large as the isoprene emissions, as a consequence of the depletion of OH and slower reaction rates compared to the BVOCs. Consequently, the effect of temperature on IEPOX-SOA is rather small, with a minuscule increase in IEPOX-SOA formed at higher temperatures at the end of the day (around 0.02 (Figure C5).

The abundance of the BVOCs leads to a build up of SVOCs in the ABL that are available for partitioning, but since partitioning to aerosol phase is generally favoured at lower temperatures, rising temperatures reduce the partitioning coefficient (Takekawa et al., 2003). Janssen et al. (2012) discussed that the partitioning efficiency of SOA had a non-linear response especially at low temperatures and high background SOA availability. At low $OA_{bg}$ concentrations and high temperatures, the partitioning coefficient is small, however there is a slight increase in SOA concentration.

A rising temperature could in principle affect the surface heat fluxes and ABL development, by increasing the vapour pressure deficit (Van Heerwaarden et al., 2009). However, we find that for a temperature increase of 2K, this effect is of minor importance (Figure C5), and the entrainment of SOA is hardly affected. Overall, a rise in temperature does not have a significant effect on modelled SOA concentration.

However, the southeastern US, in contrast to the rest of the US, has experienced cooling summer temperatures which has

been linked to either the high aerosol loading or other large scale synoptic meteorology predominant in that region (Goldstein et al., 2009; Pan et al., 2013). Cooler temperatures favour partitioning to the aerosol phase, although BVOC emissions will be lower. If the concentration of aerosol is already high, however, the low temperatures would lead to an increase in aerosol concentration. The regional cooling caused by the high aerosol concentration could further exacerbate this situation. The decrease in temperature by 1 and 2 K show that the SOA concentrations do not change much despite the decrease in available BVOCs.

This means that the cooling that has been seen over this region of the US is unlikely to have affected SOA concentrations above the region.

The radiative effect caused by the high aerosol loading means that the region is likely to stay cooler than the rest of the US (Barbaro et al., 2014; Goldstein et al., 2009), which should increase aerosol in the regions, though at cooler temperatures the BVOC emissions will be lower, which would limit the SOA formation. These radiative effects of aerosol on the surface energy

balance (Barbaro et al., 2014) are, however, not included in this work.

## 9   Conclusion

We studied the diurnal evolution of biogenic secondary organic aerosol in southeastern US, by combining the MXLCH-SOA model with observations from the SOAS campaign. By coupling the MXLCH-SOA boundary layer-chemistry model to modules




that interactively calculate surface VOC fluxes and heat fluxes,we can study the diurnal SOA evolution in the context of a tightly coupled land surface-boundary layer-SOA formation system.

An evaluation with observations shows that our model system reproduces observations of surface fluxes, tracer concentrations and boundary layer height satisfactorily. Deviations from observed mixing ratios were found for isoprene and monoterpenes

measured just above canopy. However, modelled mixing ratios of VOCs agree better with aircraft observations, which are actually more representative for the mixed layer.

We considered several mechanisms for SOA formation from isoprene and monoterpenes, though the model was limited to daytime and night-time reactions were not included. Reactive uptake of IEPOX-SOA agreed well with observations, thereby corroborating previous studies, in a case study that is tightly constrained by observations. ISOPOOH-SOA formation though

condensation is reproduced within the measurement uncertainty, although the observed peak around noon is not captured by the model. The mean isoprene SOA yield is 1.8%, which is in the lower range of values reported in literature.

Monoterpene-SOA dominates over isoprene SOA, contributing 68% to aerosol mass, with limonene having a largest contribution in the early morning (60%) which is then taken over by $\alpha$- and $\beta$-pinene. The mean monoterpene-SOA yield is 10.7%. In contrast to isoprene-SOA, there are no observed monoterpene-specific aerosol factors, so both the LO-OOA and

the MO-OOA factors may result from monoterpene-SOA formation. Our findings suggest that the more oxidised oxygenated organic aerosol (MO-OOA) could result from entrainment from the residual layer in the late morning and fast oxidation in the late afternoon, although the roles of horizontal advection, autoxidation reactions, and /or lower real MT-SOA volatility than in the VBS used here may also play a role in the observed differences. VOC oxidation by the nitrate radical contributed negligibly to SOA formation during daytime, while OH-initiated reactions dominated the SOA formation.

In a sensitivity analysis of the coupled land surface-boundary layer-SOA formation system to temperature changes, we find that the effect of increasing BVOC emissions with increasing temperatures is offset by a depletion of OH-concentrations and decrease in partitioning efficiency of SVOCs into the aerosol phase. This suggests that near-surface SOA concentrations in southeastern US are buffered against temperature changes in the region.

*Competing interests.* The authors declare that they have no conflict of interest.

*Acknowledgements.* WWH and JLJ acknowledge support from NOAA NA18OAR4310113 and EPA STAR 83587701-0. This manuscript has not been reviewed by EPA and no endorsement should be inferred.





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



## Appendix B: Model initialisation

**Table B1.** Dynamics: Initial and boundary layer conditions to reproduce the dynamical properties of June 11, 2013 from the SOAS measurement campaign, based on Su et al. (2016).

| Property | Value | Units |
|---|---|---|
| Initial boundary layer height (h) | 400 | m |
| Flow divergence factor for subsidence ($w_{sls}$) | $9 \cdot 10^{-6}$ | s$^{-1}$ |
| Surface sensible heat flux ($(\overline{w'\theta'})_s$) | * | K m s$^{-1}$ |
| Entrainment ratio ($\beta$) | 0.2 | [-] |
| Initial mixed layer potential temperature ($\langle\theta\rangle$) | 296.6 | K |
| Potential temperature lapse rate ($\gamma_\theta$) | for h<1150 m = 0.002 | K m$^{-1}$ |
| | for h>1150 m = 0.005 | K m$^{-1}$ |
| Initial potential temperature jump ($\Delta\theta$) | 1.2 | K |
| Advection of potential temperature $A_\theta$ | $5 \cdot 10^{-4}$ | K s$^{-1}$ |
| Surface moisture flux ($(\overline{w'q'})_s$) | * | g kg$^{-1}$ m s$^{-1}$ |
| Initial mixed layer specific moisture ($\langle q\rangle$) | 16.8 | g kg$^{-1}$ |
| Specific moisture lapse rate ($\gamma_q$) | -0.004 | g kg$^{-1}$ m$^{-1}$ |
| Initial specific moisture jump ($\Delta q$) | -2.0 | g kg$^{-1}$ |
| Advection of specific moisture ($A_q$) | $1.5 \cdot 10^{-4}$ | g kg$^{-1}$ s$^{-1}$ |
| Pressure | 1005.1 | Pa |

* Calculated interactively in Section 3.4



**Table B2.** Chemical Reaction Scheme. In the reaction rates, T is the absolute temperature in Kelvin and is the solar zenith angle. First-order reaction rates are in $s^{-1}$, second-order reaction rates in $cm^3$ molecule$^{-1}$ s$^{-1}$. PRODUCTS are the species which are not further evaluated in this chemical reaction scheme. The reaction scheme is derived from Janssen et al. (2013); Su et al. (2016) and new reactions adapted from Hu et al. (2016), while speciated monoterpene reactions and reactions rates are from Orlando and Tyndall (2012), Crounse et al. (2011) and Atkinson and Arey (2003). SVOCs are shown in bold, which are then distributed per bin and multiplied by the respective $\alpha$ factor.

| Number | Reaction | Reaction Rate |
|---|---|---|
| R1 | $O_3 + hv \rightarrow O^1D + O_2$ | $3.00 \cdot 10^{-5} \cdot e^{\frac{-0.575}{\cos(\chi)}}$ |
| R2 | $O^1D + H_2O \rightarrow 2OH$ | $1.63 \cdot 10^{-10} \cdot e^{\frac{60}{T}}$ |
| R3 | $O^1D + N_2 \rightarrow O_3$ | $2.15 \cdot 10^{-11} \cdot e^{\frac{110}{T}}$ |
| R4 | $O^1D + O_2 \rightarrow O_3$ | $3.30 \cdot 10^{-11} \cdot e^{\frac{55}{T}}$ |
| R5 | $NO_2 + hv \rightarrow NO + O_3$ | $1.67 \cdot 10^{-2} e^{\frac{-0.575}{\cos(\chi)}}$ |
| R6 | $CH_2O + hv \rightarrow HO_2$ | $1.47 \cdot 10^{-4} \cdot e^{\frac{-0.575}{\cos(\chi)}}$ |
| R7 | $OH + CO \rightarrow HO_2$ | $2.40 \cdot 10^{-13}$ |
| R8 | $OH + CH_4 \rightarrow CH_3O_2$ | $2.45 \cdot 10^{-12} \cdot e^{\frac{-1775}{T}}$ |
| R9 | $OH + ISO \rightarrow IRO_2$ | $2.70 \cdot 10^{-11} \cdot exp\frac{390}{T}$ |
| R10 | $OH + [MVK+MACR] \rightarrow HO_2 + CH_2O$ | $2.40 \cdot 10^{-11}$ |
| R11 | $OH + HO_2 \rightarrow H_2O + O_2$ | $4.80 \cdot 10^{-11} \cdot e^{\frac{250}{T}}$ |
| R12 | $OH + H_2O_2 \rightarrow H_2O + HO_2$ | $2.90 \cdot 10^{-12} \cdot e^{\frac{-160}{T}}$ |
| R13 | $HO_2 + O_3 \rightarrow OH + 2O_2$ | $2.03 \cdot 10^{-16} \cdot (\frac{T}{300})^{4.57} \cdot e^{\frac{693}{T}}$ |
| R14 | $HO_2 + NO \rightarrow OH + NO_2$ | $3.50 \cdot 10^{-12} \cdot e^{\frac{250}{T}}$ |
| R15 | $CH_3O_2 + NO \rightarrow HO_2 + NO_2 + CH_2O$ | $2.80 \cdot 10^{-12} \cdot e^{\frac{300}{T}}$ |
| R16 | $IRO_2 + NO \rightarrow HO_2 + NO_2 + CH_2O + 0.7[MVK+MACR]$ | $1.00 \cdot 10^{-11}$ |
| R17 | $OH + CH_2O \rightarrow HO_2$ | $5.50 \cdot 10^{-12} \cdot e^{\frac{125}{T}}$ |
| R18 | $2HO_2 \rightarrow H_2O_2 + O_2$ | * |
| R19 | $IRO_2 + HO_2 \rightarrow 0.12OH + 0.88ISOPOOH + 0.12HO_2 + 0.073MVK + PRODUC$ | $7.40 \cdot 10^{-13} \cdot exp\frac{390}{T}$ |
| R20 | $CH_3O_2 + HO_2 \rightarrow PRODUC$ | $4.10 \cdot 10^{-13} \cdot e^{\frac{750}{T}}$ |
| R21 | $OH + NO_2 \rightarrow HNO_3$ | $3.50 \cdot 10^{-12} \cdot e^{\frac{340}{T}}$ |
| R22 | $NO + O_3 \rightarrow NO_2 + O_2$ | $3.00 \cdot 10^{-12} \cdot e^{\frac{-1500}{T}}$ |
| R23 | $NO + NO_3 \rightarrow 2NO_2$ | $1.80 \cdot 10^{-11} \cdot e^{\frac{110}{T}}$ |
| R24 | $NO_2 + O_3 \rightarrow NO_3 + O_2$ | $1.40 \cdot 10^{-13} \cdot e^{\frac{-2470}{T}}$ |
| R25 | $NO_2 + NO_3 \rightarrow N_2O_5$ | ** |
| R26 | $N_2O_5 \rightarrow NO_3 + NO_2$ | *** |
| R27 | $N_2O_5 + H_2O \rightarrow 2HNO_3$ | $2.50 \cdot 10^{-22}$ |
| R28 | $N_2O_5 + 2H_2O \rightarrow 2HNO_3 + H_2O$ | $1.80 \cdot 10^{-39}$ |
| R29 | $OH + O_3 \rightarrow HO_2 + O_2$ | $1.30 \cdot 10^{-12} \cdot exp\frac{-950}{T}$ |
| R30 | $ISOPOOH + OH \rightarrow IEPOX + OH$ | $1.90 \cdot 10^{-11} \cdot exp\frac{390}{T}$ |
| R31 | $ISOPOOH + OH \rightarrow LVOC$ | $1.7 \cdot 10^{-11}$ |
| R32 | $IEPOX + OH \rightarrow PRODUC$ | $5.78 \cdot 10^{-11} \cdot exp\frac{-400}{T}$ |
| R33 | $LVOC \rightarrow 0.04ISOPOOH\text{-}SOA$ | $6.6 \cdot 10^{-3} s^{-1}$ |
| R34 | $IEPOX \rightarrow 0.11IEPOX\text{-}SOA$ | $1.54 \cdot 10^{-4} s^{-1}$ |
| R35 | $OH+SO_2 \rightarrow H_2SO_4$ | **** |
| R36 | $ISO+O_3 \rightarrow \mathbf{ISO_3}$ | $1.03 \cdot 10^{-14} \cdot exp\frac{-1995}{T}$ |





| | | |
|---|---|---|
| R37 | ISO+NO$_3$→**ISNO$_3$** | $3.15 \cdot 10^{-12} \cdot exp\frac{-450}{T}$ |
| R38 | APIN+OH→**APOH** | $1.21 \cdot 10^{-11} \cdot exp\frac{436}{T}$ |
| R39 | APIN+O$_3$→**APO$_3$** | $5.06 \cdot 10^{-16} \cdot exp\frac{-530}{T}$ |
| R40 | APIN+NO$_3$→**APNO$_3$** | $1.19 \cdot 10^{-12} \cdot exp\frac{490}{T}$ |
| R41 | BPIN+OH→**BPOH** | $1.55 \cdot 10^{-11} \cdot exp\frac{467}{T}$ |
| R42 | BPIN+O$_3$→**BPOO$_3$** | $1.2 \cdot 10^{-15} \cdot exp\frac{-1300}{T}$ |
| R43 | BPIN+NO$_3$→**BPNO$_3$** | $2.51 \cdot 10^{-12}$ |
| R44 | LIMO+OH→**LIOH** | $4.28 \cdot 10^{-11} \cdot exp\frac{401}{T}$ |
| R45 | LIMO+O$_3$→**LIO$_3$** | $2.95 \cdot 10^{-15} \cdot exp\frac{-783}{T}$ |
| R46 | LIMO+NO$_3$→**LINO$_3$** | $1.22 \cdot 10^{-11}$ |
| R47 | RO$_2$+NO →HO$_2$+NO$_2$+CH$_2$O | $8.80 \cdot 10^{-11}$ |
| R48 | RO$_2$+HO$_2$→PRODUC | $2.09 \cdot 10^{-11} \cdot exp\frac{750}{T}$ |
| R49 | RO$_2$+RO$_2$→PRODUC | $2.3 \cdot 10^{-13}$ |

$* \; k = (k1+k2)/k3; k1 = 2.21 \cdot 10^{-13} \cdot e^{\frac{600}{T}}; k2 = 1.91 \cdot 10^{-33} \cdot c_{air}; k3 = 1 + 1.4 \cdot 10^{-21} \cdot e^{\frac{2200}{T}} \cdot C_{H_2O}$

$** \; k = 0.35 \cdot (k_0 k_\infty)/(k_0 + k_\infty); k_0 = 3.61 \cdot 10^{-30} \cdot (\frac{T}{300})^{-4.1} \cdot c_{N_2}; k_\infty = 1.91 \cdot 10^{-12} \cdot (\frac{T}{300})^{0.2}$

$*** \; k = 0.35 \cdot (k_0 k_\infty)/(k_0 + k_\infty); k_0 = 1.31 \cdot 10^{-3} \cdot (\frac{T}{300})^{-3.5} \cdot e^{\frac{-11000}{T}} \cdot c_{N_2}; k_\infty = 9.71 \cdot 10^{14} \cdot (\frac{T}{300})^{0.1} \cdot e^{\frac{-11080}{T}}$




**Table B3.** Initial mixing ratio in the boundary layer and FT surface emission/deposition fluxes of reactants based on Su et al. (2016). Gas-phase chemistry conditions are based on ground observations at SEARCH site, flux tower observations at the AABC tower and aircraft observations (WASP system and NCAR-130 flight) and then averaged for 5, 6, 8, 10-13 June (Su et al., 2016). Observations for secondary organic aerosol are from the Aerosol Mass Spectrometer on the SEARCH ground site and a SENEX flight on 11 June. Species with 0 initial concentrations and emissions are not included in the table. The SVOCs have a 0 initial concentrations, but have a deposition velocity of 0.024 m/s (not mentioned in the table).

| Species | Initial mixing ratio (ppb) | | Emission/Deposition |
| :---: | :---: | :---: | :---: |
| | ABL mixing ratio | FT mixing ratio | (ppb m/s) |
| $O_3$ | 14.0 | 51 | 0.023* |
| NO | 0.1 | 0.05 | $-0.005\sin(\frac{\pi t}{t_d})$ |
| $NO_2$ | 0.5 | 0.08 | $0.005\sin(\frac{\pi t}{t_d})$ |
| HCHO | 2.0 | 1.1 | 0.0 |
| ISO | 0.6 | 0.0 | ** |
| MVK+MACR | 0.6 | 0.6 | 0.024* |
| $OA_{BG}$ *** | 0.32 | 0.15 | 0.0 |
| ISOPOOH | 0.0 | 0.0 | 0.03* |
| IEPOX | 0.0 | 0.0 | 0.03* |
| IEPOX-SOA | 0.06 | 0.06 | 0.0 |
| ISOPOOH-SOA | 0.014 | 0.014 | 0.0 |
| APIN | 0.45 | 0 | 0.45×** |
| BPIN | 0.45 | 0 | 0.45×** |
| LIMO | 0.1 | 0 | 0.1×** |

* Dry deposition velocity in m s[-1]

**Interactively calculated in Section 3.3

*** The OA is converted to $\mu$ g m[-1] in the model using a molecular weight of 250 [g mol[-1]] multiplied by the pressure [Pa], divided by the gas constant R [8.3145 J mol[-1] K[-1]] and the potential temperature [K] at half the boundary layer height, all multiplied by 0.001 (to convert it to $\mu$m)





**Table B4.** Advanced surface variables: plant and soil initial and boundary layer conditions to study the effect of a coupled land-atmosphere scheme. The plant scheme has been taken from Van Heerwaarden et al. (2009)'s value for the broad leaf trees (deciduous forests) sand loam soil, with some some observations taken from the Integrated Surface Flux System measurements taken at the AABC flux tower.

| Property | Value | Units |
|---|---|---|
| Initial surface (skin) temperature ($T_s$) | 298.6 | K |
| Soil moisture (wg) | 0.29 | $m^3 m^{-3}$ |
| Soil moisture deeper soil layer (w2) | 0.22 | $m^3 m^{-3}$ |
| Wilting point (wwilt) | 0.171 | $m^3\ m^{-3}$ |
| Volumetric water content field capacity (wfc) | 0.323 | $m^3\ m^{-3}$ |
| Saturated volumetric water content (wsat) | 0.472 | $m^3\ m^{-3}$ |
| CL* parameter a | 0.219 | [-] |
| CL* parameter b | 4.9 | [-] |
| CL* parameter c | 4.0 | [-] |
| Coefficient force term moisture (C1sat) | 0.132 | [-] |
| Coefficient restore term moisture (C2ref) | 1.8 | [-] |
| VPD correction factor for $r_s$ (gD) | 0.03 | [-] |
| Transpiration resistance ($r_{s;min}$) | 200 | $s\ m^{-1}$ |
| Soil transpiration resistance ($r_{soil;min}$) | 20 | $s\ m^{-1}$ |
| Leaf Area Index (LAI) | 5 | $m^2 m^{-2}$ |
| Vegetation fraction $c_{veg}$ | 0.9 | [-] |
| Initial temperature top soil layer | 294.6 | K |
| Temperature deeper soil layer (T2) | 293.6 | K |
| Thermal conductivity skin layer divided by depth ($\Lambda$) | 20 | $W\ m^{-2}\ K^{-1}$ |
| Roughness length momentum ($z_{om}$) | 2.0 | m |
| Roughness length heat ($z_{oh}$) | 2.0 | m |

*Clapp and Hornberger retention curve parameter



**Table B5.** MEGAN parameters and values used in the mixed layer model

| Property | Value | Units |
|---|---|---|
| Base Emission Rate, Isoprene $\epsilon_{Iso}$ | 7900(=2.11) | $\mu g m^{-2} hr^{-1} (s^{-1})$ |
| Production and loss rate, Isoprene $\rho_{Iso}$ | 0.96 | [-] |
| Emission activity factor, leaf age $\gamma_{Age}$ | 1 | [-] |
| Emission activity factor, soil moisture $\gamma_{SM}$ | 1 | [-] |
| Soil moisture ($\theta$) | 0.40 | $m^3 m^{-3}$ |
| Wilting point ($\theta_w$) | 0.29 | $m^3 m^{-3}$ |
| Leaf Area Index LAI | 5 | $m^3 m^{-3}$ |
| $P_{ac}$* | (PAR** $\times$ 4.766) | $\mu mol m^{-2} s^{-1}$ |
| $P_{daily}$*** | 500 | $\mu mol m^{-2} s^{-1}$ |
| Empirical coefficient $C_{T1}$ | 80 | [-] |
| Empirical coefficient $C_{T2}$ | 200 | [-] |
| Daily average air temperature $T_{daily}$ | 298 | K |
| Base Emission Rate, Monoterpene $\epsilon_{MT}$ | 860* (=0.24) | $\mu g m^{-2} hr^{-1} (s^{-1})$ |
| Production and loss rate, Isoprene $\rho_{MT}$ | 1 | [-] |
| Empirical coefficient $\beta_{MT}$ | 0.13 | $K^{-1}$ |
| Skin temperature $T_s$ | 298 (initial value) | K |
| Reference temperature $T_{ref}$ | 303 | K |

*Above canopy photosynthetically photon density flux

**Photosynthetically active radiation in $W m^{-2}$

*** Daily mean of above canopy photosynthetically photon density flux





**Table B6.** Stoichiometric coefficients for different volatility bins for precursors: $\alpha$-pinene (APIN), $\beta$-pinene (BPIN), limonene (LIMO) and isoprene (ISO) and depending on the oxidant (OH, $NO_3$ and $O_3$) at 298 K. ISO+OH is not considered as this is included in the reactive uptake and condensation pathways, and ISO+NO pathway is not considered due to the low $NO_x$ availability in this region. Saturation concentrations, $C_i^*$, are in $\mu g\ m^{-3}$, are based on Pye et al. (2010).

| i | 1 | 2 | 3 | 4 |
|---|---|---|---|---|
| Effective saturation concentration, $C_i^*$ | 0.1 | 1 | 10 | 100 |
| APIN(OH+$O_3$),low-$NO_x$ | 0.08 | 0.019 | 0.18 | 0.03 |
| APIN(OH+$O_3$),high-$NO_x$ | 0.04 | 0.0095 | 0.09 | 0.015 |
| APIN($NO_3$) | 0 | 0 | 0 | 0 |
| BPIN(OH+$O_3$),low-$NO_x$ | 0.08 | 0.019 | 0.18 | 0.03 |
| BPIN(OH+$O_3$),high-$NO_x$ | 0.04 | 0.0095 | 0.09 | 0.015 |
| BPIN($NO_3$) | 0 | 0 | 0.321 | 1.083 |
| LIMO(OH+$O_3$),low-$NO_x$ | 0 | 0.366 | 0.321 | 0.817 |
| LIMO(OH+$O_3$),high-$NO_x$ | 0 | 0.474 | 0.117 | 1.419 |
| LIMO($NO_3$) | 0 | 0.000 | 0.321 | 1.083 |
| ISO($O_3$),low-$NO_x$ | - | 0.031 | 0.000 | 0.095 |
| ISO($O_3$),high-$NO_x$ | - | 0.001 | 0.023 | 0.015 |
| ISO($NO_3$) | - | 0 | 0.217 | 0.092 |





**Appendix B: Interactive isoprene and monoterpene emissions calculations**

This parameterization is based on Guenther et al. (2006). In this model, the emissions, E, of isoprene, and other BVOCs, are parametrized by:

$$E = [\epsilon][\gamma][\rho] \tag{B1}$$

Here, $[\epsilon]$ are the base emissions in $\mu$ g m$^{-2}$ hr$^{-1}$ of compound, while $\rho$ accounts for the production and loss of the BVOC within canopy, which, for isoprene, is set to 0.96 (Guenther et al., 2006). The base emission rates are dependent on the plant functional type, and since we are over a broadleaf forest the emission rate for isoprene is set at 3000 $\mu g m^{-2} h^{-1}$ (=0.83 $\mu g m^{-2} s^{-1}$); though it is low for a broad-leaf area it is used as it is able to reproduce the isoprene mixing ratio observations. $\gamma$ (dimensionless) is an emission activity factor and represents variation in emissions due to changes from standard conditions. It is derived for

isoprene per:

$$\gamma = \gamma_{CE} \times \gamma_{Age} \times \gamma_{SM} \tag{B2}$$

$\gamma$ is a lumped correction factor (Wang et al., 2017); it takes into account effect of the canopy environment $\gamma_{CE}$, the leaf age $\gamma_{Age}$ and soil moisture $\gamma_{SM}$.

A constant value for $\gamma_{Age}$ is used ($\gamma_{Age} = 1$). In order to calculate the $\gamma_{CE}$, we utilise the parameterized canopy environment

emission activity (PCEEA) algorithm. This is calculated by:

$$\gamma_{CE} = \gamma_T \times \gamma_P \times \gamma_{LAI} \tag{B3}$$

The parameterized $\gamma$'s are activity factors that are related to variations of temperature (t), light and the leaf area index (LAI) (Guenther et al., 2006); $\gamma_T$ is temperature dependent, $\gamma_{LAI}$ depends on the leaf area index while $\gamma_P$ represents the leaf-level photosynthetic photon flux density (PPFD), with units in $\mu$ mol m$^{-2}$ s$^{-1}$. The PPFD is related to the photosynthetically active

radiation (PAR) (Guenther et al., 2006). PAR is the radiation that organisms can use for photosynthesis and in our model framework, the PAR depends on the incoming solar radiation.

Isoprene emissions respond to changes in PPFD at canopy-level by:

$$\gamma_P = 0 \qquad a < 0, a > 180 \tag{B4a}$$

$$\gamma_P = \sin(a)[2.46(1 + 0.0005 \cdot (P_{daily} - 400))\phi \cdot 0.9\phi^2] \qquad 0 < a < 180 \tag{B4b}$$





where $a$ is the solar angle (calculated by subtracting the zenith angle from 90 degrees) in degrees. $P_{daily}$ is related to the PAR (multiplied by 4.766 to convert it from W m$^{-2}$ to $\mu$ mol m$^{-2}$ s$^{-1}$) and represents the daily mean of the above canopy PPFD, and $\phi$ is the transmission of the above canopy PPFD which is non-dimensional (Guenther et al., 2006) and approximated by:

$$\phi = P_{ac}/(\sin(a)P_{toa}) \tag{B5}$$

The $P_{ac}$, the above canopy PPFD, is also approximated from PAR multiplied by a conversion factor (4.766). $P_{toa}$, the top of the atmosphere PPFD (Guenther et al., 2006), depends on the day of the year (DOY).

$$P_{toa} = 3000 + 99 \cdot \cos(2 \cdot 3.14 \cdot (DOY - 10)/365) \tag{B6}$$

The response of isoprene emissions to temperature is calculated by:

$$\gamma_T = E_{opt} \times [C_{T2} \times exp(C_{T1} \times x/(C_{T2} - C_{T1} \times (1 - exp(C_{T2} \times x))))] \tag{B7}$$

Here $x = [(1/T_{opt}) - (1/T]/0.00831$, $C_{T1}$ (=80) and $C_{T2}$ (=200) are empirically derived coefficients, $T_{opt}$ is the optimal temperature at which $E_{opt}$ is calculated (Guenther et al., 2006).

$$T_{opt} = 313 + (0.6 \times (T_{daily} - 297) \tag{B8}$$

$$E_{opt} = 1.75 \times exp(0.08 \times (T_{daily} - 297)) \tag{B9}$$

where $T_{daily}$ is the representative daily average air temperature at canopy level for the modelling period (K) which is set to
298 K, based on surrounding temperature measured at the SOAS campaign site. Lastly, for canopy-level, the isoprene emission dependence on the leaf area index (LAI in $m^3 m^{-3}$) is estimated by:

$$\gamma_{LAI} = 0.49LAI/[(1 + 0.2LAI^2)^{0.5}] \tag{B10}$$

The last $\gamma$ factor, $\gamma_{SM}$, is 1 if the soil moisture, $\theta$ is greater than $\theta_l$; 0 if $\theta$ is less than the wilting point, $\theta_w$, and $(\theta - \theta_w)/\Delta\theta_l$ if $\theta_w$ is less than $\theta$ which is less than $\theta_l$; $\Delta\theta_l$ is an empirical parameter equalling 0.06 (Guenther et al., 2006).
For the monoterpene flux, in Equation (B1), the $\rho = 1$, $\epsilon = 850$ $\mu$ g m$^{-2}$ h$^{-1}$ to fit the monoterpene mixing ratio observations and $\gamma$ is given by:

$$\gamma = \gamma_{CE} \times \gamma_{SM} \tag{B11}$$





As above, it is determined by the canopy emission activity factor and the soil moisture emission activity factor. The soil moisture emission factor is 1. The canopy emission activity factor is calculated by:

$$\gamma_{CE} = \gamma_{LAI} \times \gamma_T \tag{B12}$$

which depends on the LAI emission activity factor, $\gamma_{LAI}$, and the temperature emission activity factor, $\gamma_T$. The $\gamma_{LAI}$ is also 1, however the temperature emission activity factor is approximated by:

$$\gamma_T = exp(\beta_{MT} \times (T_s - T_{ref})) \tag{B13}$$

Here $\beta_{MT}$ is the beta (an empirical coefficient) for monoterpene, set at 0.1 $K^{-1}$ (Guenther et al., 2006), $T_s$ is the skin temperature and $T_{ref}$ is the reference temperature for BVOC Base Emission Rate (K) and equals 303 K.





## Appendix C:  Supporting Figures



**Figure C1.** (a) Ozone, (b) nitrogen oxide, (c) nitrogen dioxide and (d) hydroxide (OH) mixing ratio measured (blue) at SOAS super site versus modelled by MXLCH model (red) over the SOAS super site during the SOAS measurement campaign for the days 5, 6, 8, 10-13 June, 2013.




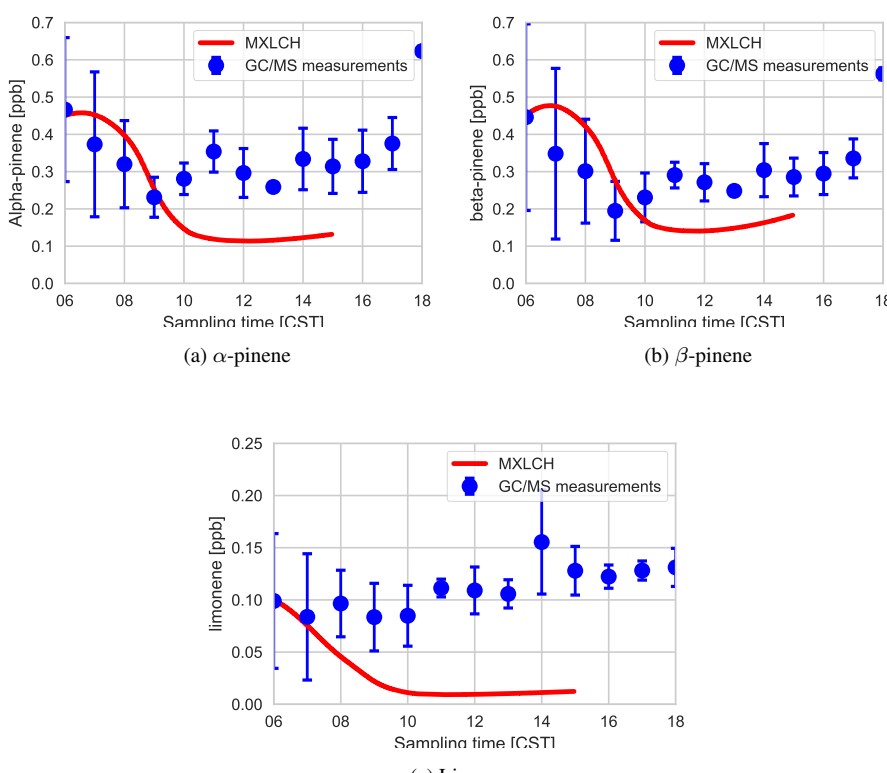

**Figure C2.** Mixing ratios of (a) alpha-pinene, (b) beta-pinene and (c) limonene measured (blue) by gas chromatography mass spectrometry over the SOAS super site tower (20 m, above canopy), averaged over the 5-13 June, 2013 versus the mixing ratios modelled (red) in the MXLCH-SOA model.



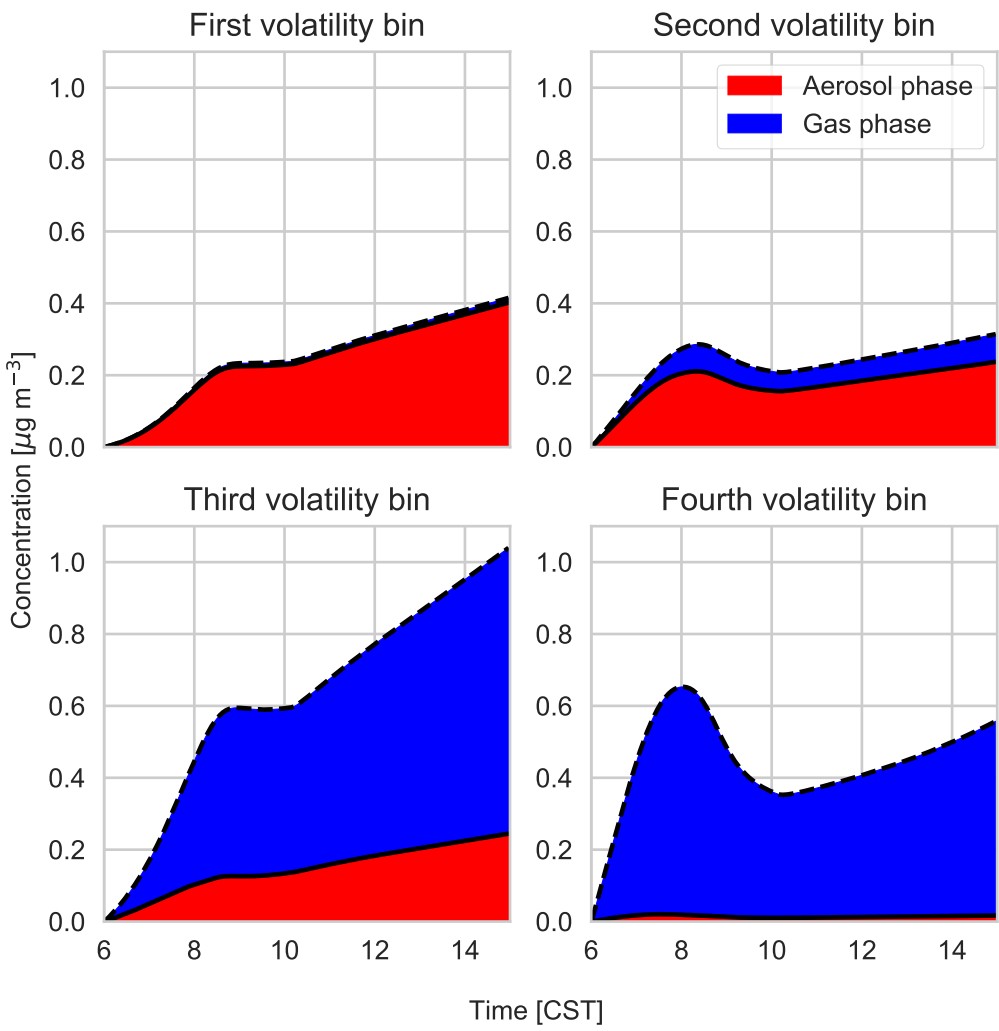

**Figure C3.** Gas-particle partitioning products per volatility bin, with red indicating amount of SVOC in aerosol phase, versus gas-phase (blue).





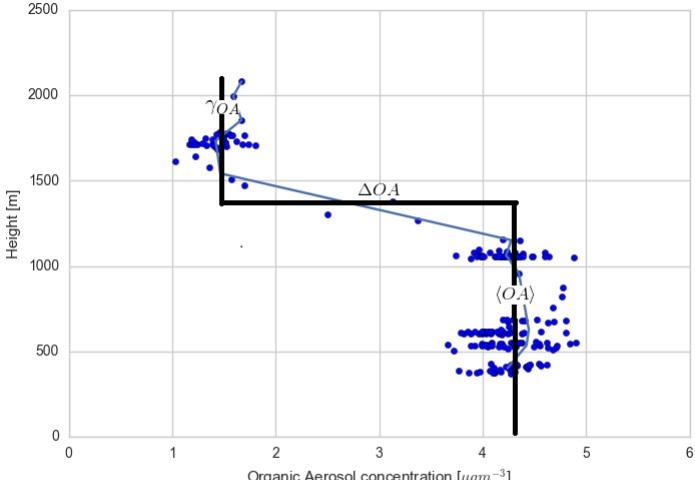

**Figure C4.** Measured vertical profile of organic aerosol (blue dots) taken during the SENEX campaign above the SOAS campaign sites on June 11, 2013 at 14:00 CST, averaged for different heights (blue line) and overlaid with a typical convective boundary layer vertical profile; a mixed layer represented by a bulk value ($\langle OA \rangle$), a sharp discontinuity in the inversion layer ($\Delta OA$) and a value in the free troposphere ($\gamma_{OA}$)

.

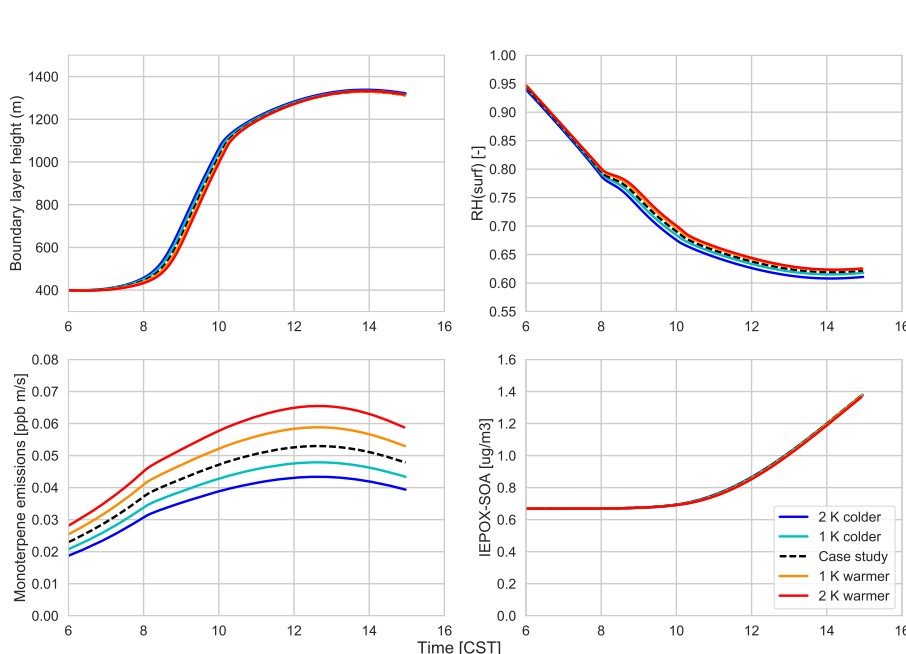

**Figure C5.** Effect of temperature on boundary layer height (top-left), relative humidity (top-right), monoterpene emissions (bottom-left) and IEPOX-SOA (bottom-right)




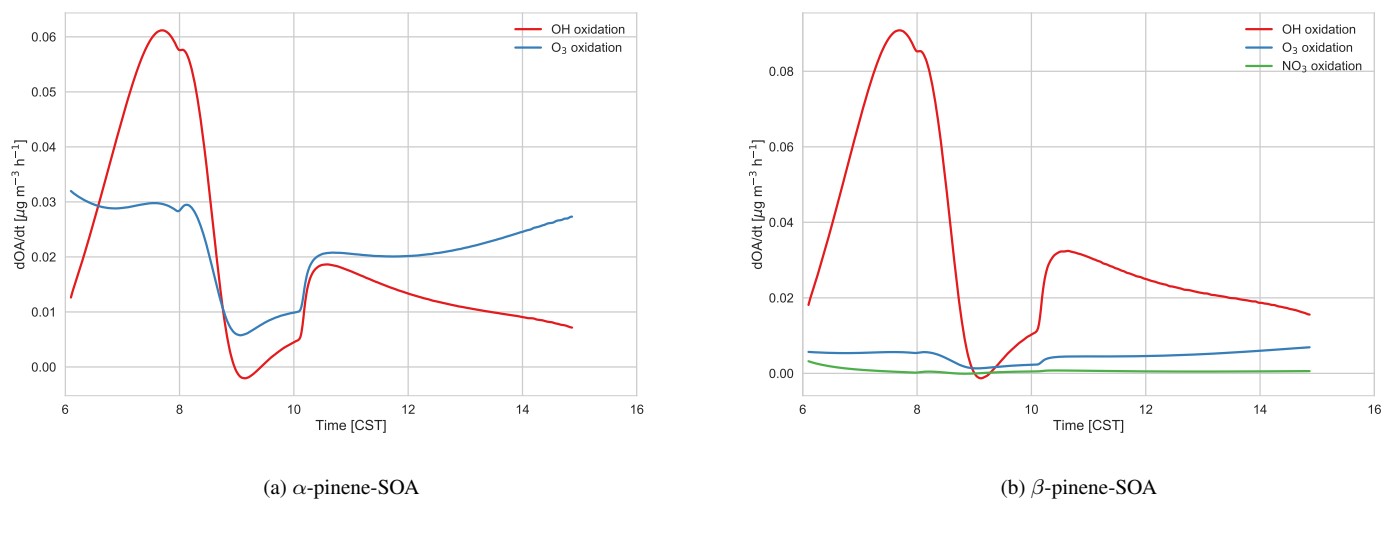

(a) $\alpha$-pinene-SOA

(b) $\beta$-pinene-SOA

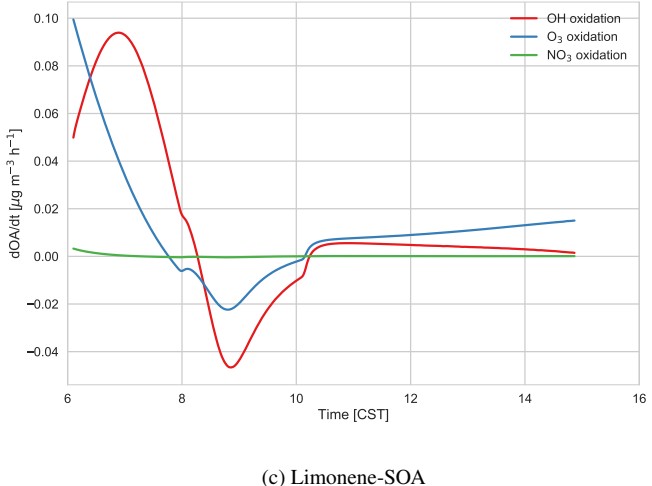

(c) Limonene-SOA

**Figure C6.** (a) $\alpha$-pinene SOA, (b) $\beta$-pinene SOA and (c) limonene-SOA divided by oxidant contribution (OH, $O_3$ and $NO_3$). $\alpha$-pinene SOA has no contribution from $NO_3$ (See Table B6).