# Peer review of "Biogenic emissions and land-atmosphere interactions as drivers of the daytime evolution of secondary organic aerosol in the southeastern US"

_Atmospheric Chemistry and Physics, 2018_

## Referee Comment (RC1) · Anonymous Referee #1 · 6 Sep 2018

The paper by Nagori et al. summarizes results from a coupled land-atmosphere chemical modeling study to predict daytime SOA formation and evolution in SEUS. The model is constrained by observations of trace gases and organic aerosol measurements from different platforms (ground and airborne). SOA formation sensitivity to future temperature changes in the range of +/-2 degrees was also tested. Results indicate that daytime OH oxidation of monoterpenes and isoprene are the most important sources of SOA. Sensitivity runs show that future SOA formation is relatively insensitive to warming or cooling in the region (+/-2 degrees). There have been a lot of studies on SOA formation in SE-US. The unique aspect of this paper is in using a coupled-land/atmosphere chemistry model to consider daytime evolution of SOA. The

paper is overall well-written and figures/tables are good quality and clear.

Technical comments: My major concern is regarding the nighttime oxidation of monoterpenes. The paper as written is very confusing to know if NO3 chemistry of BVOCs at night was modeled or not. I understand that the reported diurnal profiles are from 6 am to 6 pm, but without a full simulation of nighttime chemistry, starting values of SOA at 6 am cannot be correct. I think the title needs to indicate specifically that the focus is the evolution of SOA during 'daytime'. Also, there was no indication of the chemistry of sesquiterpene VOCs. I think at least some discussion of why such VOCs were not considered should be provided. Please see below my other comments/concerns/questions. I would like to see the authors' response to my comments before the paper is accepted for publication.

P3, L5: nighttime SOA formation. Many of the papers that the authors cite have indicated a significant effect of NO3 chemistry on SOA formation in the region, so why does this study not include such chemistry? Contradicting to the sentence on P3, L5, on P6, L27 and 31 (also Table B2) authors indicate that NO3 oxidation of BVOCs were considered, so how can these two statements be both true? Does it mean spin up time includes nighttime chemistry so what you start at sunup includes such products? Related to this, on P16, L28: Why isn't NO3 oxidation to form SOA in the early morning hours significant in this study? It's not surprising that daytime contribution is almost zero, but early morning should be showing this contribution (according to other studies for the same region). P23, L8: Again, do you mean NO3 reactions were not included only during the day or not even at night?

P7, L6: It was mentioned long range transport wasn't modeled. If your model includes only local reactions/production of SOA, wouldn't you be able to estimate how much local production of MO-OOA there was if entrainment is set to zero?

P8, L6-7: since uptake coefficient of IEPOX depends on inorganic composition of SOA, how is that estimated? It seems that only OA is simulated in the model.

[Figure]

P9, section 3.4. Is there a direct input from the coupled land-surface mode to MEGAN to include the effects of modeled soil moisture variabilities on monoterpene emissions or is the soil moisture emission activity factor insensitive to the range of soil moisture changes in this environment?

P15, L6: it's mentioned that ISOPOOH SOA peak is not captured in the model, but no explanation is provided.

P18, L5: How is the entrainment curve in Fig. 8 determined?

P19, L2: Monoterpenes have a continuous source as long as there is light and warm temperature, so I don't think the statement in this sentence that most of monoterpenes have reacted in the morning makes sense. Please clarify.

P19, L16: I would clarify that OH is the most important daytime oxidant during for SOA formation.

P20, L5-12: looking at the diurnal pattern of LO-OOA, it actually appears that LO-OOA concentration decreases in early morning and then increases, suggesting that LO-OOA is locally produced. This trend is in contrast to the trend in MO-OOA, so doesn't this suggest that MO-OOA is actually more regional and present at higher conc. in the RL than the BL? This is mentioned in the conclusions, but not in this section. I think both of these trends are worth highlighting again here.

P21, L5: Does the impact of higher air temperature on soil moisture content and response of vegetation's BVOC emissions to changes in soil moisture content also represented in the climate sensitivity runs? If not, how would such results be different?

Editorial comments:

P2, L20: Another recent paper discussing importance of monoterpenes in SE-US by Xu et al. (ACP, 2018) should also be referenced. (Xu, L., Pye, H. O. T., He, J., Chen, Y., Murphy, B. N., and Ng, N. L.: Experimental and model estimates of the contributions from biogenic monoterpenes and sesquiterpenes to secondary organic

aerosol in the southeastern United States, Atmos. Chem. Phys., 18, 12613-12637, https://doi.org/10.5194/acp-18-12613-2018, 2018.)

P5, L6: add "...(model results) are shown..."

P5, L15: change "sides" to "sites"

P5, L25: remove one "been"

P7. Numbering for tables and figures seems not to be in the right order

P7, L26: "an early morning..."

P8, L17: I would remove "the" of "the emissions..."

P9, L9: consider changing "," to ";" or start a new sentence

P19, L30: "...leads to overestimated values compared to ..."
* * *

---

## Referee Comment (RC2) · Anonymous Referee #2 · 18 Sep 2018

The authors use existing parameterisations to update a coupled land-atmosphere model to represent biogenic secondary organic aerosol (SOA) formation at a site in the southeast US. The paper is clearly written and the figures and tables are well presented. The topic is also appropriate for ACP, but the justification for the study is weak.

The authors use mechanisms and processes that have already been applied to regional and global chemistry transport models (CTMs) (Pye et al., 2010, 2013; Marais et al., 2016) and used to describe the processes that contribute to biogenic SOA formation. It's not apparent why a land model is needed to improve the SOA simulation or that conclusions from this study couldn't be derived without the land model. There is

also no evidence or reference to past studies to support the need for the land model. What happens to the simulations when input from the land model is prescribed rather than modelled explicitly?

The model that the authors use also only simulates atmospheric composition for a limited time period during the day. It's not clear why this is the case and what effect this has on simulation of biogenic SOA. The output from the model also gives the impression that there is something faulty with the model, especially in Figure 5 (b) where it appears simulated isoprene mixing ratios would continue to increase beyond 3pm (when the last output is obtained from the model) and so far exceed the measurements.

There are also inappropriate references provided to indicate the source of parameterizations and variables for the biogenic SOA formation mechanisms. For example, the authors quote Hu et al. (2016) numerous times (P2 L17, P2 L19, and throughout Section 3.2, but the variables and parameterisations are not original to that study.

References:

Marais et al., doi:10.5194/acp-16-1603-2016, 2016. Pye et al., doi:10.5194/acp-10-11261-2010, 2010. Pye et al., doi:10.1021/es402106h, 2013.
* * *

---

## Author Comment (AC1) · 9 Nov 2018

**Response to reviewers for the paper: "Biogenic emissions and land-atmosphere interactions as drivers of the diurnal evolution of secondary organic aerosol in the southeastern US." Juhi Nagori, Ruud H. H. Janssen, Juliane L. Fry, Maarten Krol, Jose L. Jimenez, Weiwei Hu, and Jordi Vilà-Guerau de Arellano.**

We thank the reviewers for their comments on our paper. To guide the review process we have copied the reviewer comments in black text. Our responses are in regular blue font. We have responded to all the referee comments and made alterations to our paper (**in bold text**).

**Anonymous Referee #1**

R1.0. Overview: The paper by Nagori et al. summarizes results from a coupled land-atmosphere chemical modeling study to predict daytime SOA formation and evolution in SEUS. The model is constrained by observations of trace gases and organic aerosol measurements from different platforms (ground and airborne). SOA formation sensitivity to future temperature changes in the range of +/-2 degrees was also tested. Results indicate that daytime OH oxidation of monoterpenes and isoprene are the most important sources of SOA. Sensitivity runs show that future SOA formation is relatively insensitive to warming or cooling in the region (+/-2 degrees). There have been a lot of studies on SOA formation in SE-US. The unique aspect of this paper is in using a coupled-land/atmosphere chemistry model to consider daytime evolution of SOA. The paper is overall well-written and figures/tables are good quality and clear.

R1.1. Technical comments: My major concern is regarding the nighttime oxidation of monoterpenes. The paper as written is very confusing to know if $NO_3$ chemistry of BVOCs at night was modeled or not. I understand that the reported diurnal profiles are from 6 am to 6 pm, but without a full simulation of nighttime chemistry, starting values of SOA at 6 am cannot be correct. I think the title needs to indicate specifically that the focus is the evolution of SOA during 'daytime'.

The focus of our study is the simultaneous evolution of boundary layer, chemical, and aerosol parameters during daytime. In order to understand the diurnal evolution of SOA, the evolution of gas-phase chemistry and aerosol needs to be correctly represented, for which the boundary layer dynamics and entrainment of chemicals need to be modelled well (We discuss this on P2 L30-32). This is often a challenge for global and regional chemistry models,which may lead to inaccurate aerosol concentrations in models. The MXL model is able to correctly represent the combined effects of these processes. However as its validity is limited to convective conditions, we are only able to focus on the daytime evolution of the aforementioned processes.

We initialise the model with a bulk OA background concentration in the early morning, which is based on the measurements during SOAS. We treat this bulk OA as if there is only one generic OA mass present in the early morning (hence, not divided per oxidant/BVOC) that is strongly oxidised and non-volatile. Hence the nighttime chemistry is not modelled explicitly and the SOA sources we use are limited to the daytime only.

NO$_3$ is discussed in detail below in response to comment R1.3.

We change the title of the paper to clarify that the model is done on a sub-diurnal scale from daytime sources. The paper will now be titled: "Biogenic emissions and land-atmosphere interactions as drivers of the **daytime** evolution of secondary organic aerosol in the southeastern US."

We have also clarified the text in several places to address this point:

On P1 L8 "boundary layer (ABL)**, to gain insight in the drivers of daytime evolution of biogenic SOA.**"

On P2 L28: "Here, we study the formation of SOA from biogenic emissions **(specifically from daytime sources)** and the **SOA** diurnal evolution …"

On P2 L30-32: The diurnal SOA evolution is driven by ABL dynamics and the interaction between the ABL and the free troposphere (FT), as well as by emissions, chemical transformations and subsequent partitioning into the aerosol phase (Janssen et al., 2012, 2013). **The ABL dynamics are often a challenge to represent in global and regional chemistry models and hence,** in order to encompass [...] accurately represent the diurnal evolution of SOA **and** BVOC concentrations

On P5 L33 "representative (**sub-**diurnal)..."

P22 L33: "We studied the diurnal evolution of biogenic secondary organic aerosol**, formed from daytime sources,** in southeastern US.

R1.2. Also, there was no indication of the chemistry of sesquiterpene VOCs. I think at least some discussion of why such VOCs were not considered should be provided.

Sesquiterpenes were recently estimated as being responsible for only 3% of the OA during SOAS, compared to 45% for monoterpenes and 18% for isoprene (Hu et al., 2015; Marais et al., 2016; Zhang et al., 2018).  Hence, we focused on the two observed BVOC types that have been reported to make major contributions to SOA during SOAS, namely isoprene and monoterpenes.  Consistent with the SOAS findings, Yee et al., 2018 found a contribution of only around 0.5-5% of sesquiterpenes to SOA mass in the Amazonia, despite their high reactivity. Sesquiterpenes are very reactive and those contributions could be underestimated. However, without further information that would suggest a larger importance in the SE US, we conclude that the sesquiterpene contribution to SOA at SOAS is minor, and hence the chemistry of these sesquiterpene VOCs has not been considered. The following text has been added to the paper to clarify this point:

P2 L22: **"Sesquiterpene oxidation and the resulting SOA are not included due to its small contribution to SOA during SOAS (3%, compared to ~45% for monoterpenes and ~18% for isoprene; Hu et al., 2015; Marais et al., 2016; Zhang et al., 2018). Sesquiterpenes are very reactive and those contributions**

**could be underestimated. However, without further information that would suggest a larger importance in the SE US, we did not include sesquiterpenes in the current study."**

Please see below my other comments/concerns/questions. I would like to see the authors' response to my comments before the paper is accepted for publication.

R1.3. P3, L5: nighttime SOA formation. Many of the papers that the authors cite have indicated a significant effect of NO3 chemistry on SOA formation in the region, so why does this study not include such chemistry? Contradicting to the sentence on P3, L5, on P6, L27 and 31 (also Table B2) authors indicate that NO3 oxidation of BVOCs were considered, so how can these two statements be both true? Does it mean spin up time includes nighttime chemistry so what you start at sunup includes such products? Related to this, on P16, L28: Why isn't NO3 oxidation to form SOA in the early morning hours significant in this study? It's not surprising that daytime contribution is almost zero, but early morning should be showing this contribution (according to other studies for the same region). P23, L8: Again, do you mean NO3 reactions were not included only during the day or not even at night?

As mentioned in the response to comment R1.1, we only look at the chemistry that leads to formation of SOA during daytime, with the initial concentration of SOA at the start of the day being constrained by observations. The $NO_3$ + BVOC reactions and SOA formation are included in the model since Ayres et al., (2015) mentioned that they might still have a contribution during daytime. NO and $NO_2$ are initialized at 6am, based on observed mixing ratios (see Figure C1), which constrains the early morning $NO_x$ chemistry and subsequent SOA formation through the $NO_3$ pathway .

Hence, the NO, $NO_2$, $NO_3$ and $N_2O_5$ chemistry is included in the model, but since the $NO_3$ and $N_2O_5$ mixing ratios are rather low in the early morning (0.03 and 0.015 ppt, resp.) it does not lead to significant SOA formation.

Even though the nighttime chemistry is not simulated explicitly, the SOA initial conditions are prescribed from observations, which means that the SOA that is formed from night time reactions is accounted for. However, since the initial concentration is a bulk SOA value we cannot separate the NO3-oxidation contribution out from this early morning SOA term.

We have modified the paper text as follows to address these points:

In text: P9L 27 **"Early morning $NO_x$ chemistry and subsequent SOA formation is constrained by the initialization of NO and $NO_2$ mixing ratios at 6am, based on observed mixing ratios (see Figure C1)"**

P6, L29-30 we deleted 'As the model starts at sun-up and ends in the mid-afternoon, we do not include night-time reactions, though we start with a background SOA that is observed at sun-up.'

P9, L28 we added '**Since the model is initialized at sunrise, it does not explicitly account for nighttime SOA formation, but the effect of NO$_3$-initiated nighttime SOA formation is included in the value of the prescribed bulk SOA concentration.**'

P19, L14/15 "The isoprene + O$_3$ and NO$_3$ pathways lead to a negligible amount of SOA formed in our model, **even in the early morning. The early morning NO$_x$ chemistry and subsequent SOA formation are constrained through the observed NO and NO$_2$ initial mixing ratios. Since the resulting NO$_3$ (and N$_2$O$_5$) mixing ratios are very small, the NO$_3$-initiated SOA formation is negligible.**"

R1.4. P7, L6: It was mentioned long range transport wasn't modeled. If your model includes only local reactions/production of SOA, wouldn't you be able to estimate how much local production of MO-OOA there was if entrainment is set to zero?

We do somewhat discuss this in Section 7, with sensitivity analyses of different RL concentrations. Our initial approach was to first set initial and boundary conditions that were constrained by observations, and then run the model assuming no long range transport in order to minimize uncertainties. We then compared these results with the observed diurnal variability. Then we considered adding advective terms, if that seemed necessary.

We did perform the no-entrainment experiment and can draw some conclusions about long range transport and MO-OOA, but it is still likely that some of it is rapidly formed via autoxidation reactions, which still leaves some uncertainty in the amount that is transported versus formed locally.

To clarify we have added the following text:

P7 L5/6 "However, it is uncertain how much of the aged MO-OOA is locally formed versus advected via long-range transport, **and we use a simulation with no entrainment in an attempt to separate these effects**. **This is explored in Section 7, where different residual layer SOA concentrations are applied to explore their effect on the diurnal evolution of SOA in the ABL.**"

P9, L32 we deleted 'We use the range of these observations as constraints for the numerical experiments.' and inserted '**In addition to the base case in which SOA in the RL was initiated at 1.5 ug m$^{-3}$, we also run simulations in which we initiated it at 1 and 1.8 ug m$^{-3}$, respectively, which encompasses the range of observed SOA concentrations above the ABL. Further, we included a scenario in which SOA concentrations in the ABL and RL were initialised with uniform values. The latter scenario is then used to estimate the contribution of long-range transport versus local formation of MO-OOA.**'

P20 L7: we omit "it seems like"

> P20 L10: "**The more oxidized oxygenated organic aerosol (MO-OOA) could result from entrainment from the RL though** available measurements show that the OA […] **so** not all the aged MO-OOA can be explained by this process…"

R.1.5. P8, L6-7: since uptake coefficient of IEPOX depends on inorganic composition of SOA, how is that estimated? It seems that only OA is simulated in the model.

The IEPOX uptake coefficient is modelled using the methods from Hu et al. (2016) and Gaston et al. (2014) which take into account the inorganic composition of SOA. The aerosol composition used in the calculation of the uptake coefficient is prescribed based on the ambient observations during SOAS. From the supplement of Hu et al., (2016): "The other main parameters input into the model include temperature (298 K), mass accommodation coefficient (0.1) (Gaston et al., 2014), particle radius (140 nm, Fig. S12c), *concentration of nucleophiles including $SO_4^{2-}$ and $NO_3^-$* (~ 0.05±0.1 M and 0.23±0.23 M in ambient and OFR respectively) and $HSO_4^-$ concentration (0.59±2.4 M and 0.09±0.12 M) obtained from the output of E-AIM II model…. *the radius of the inorganic core* was input in the model, which was estimated to be 100 nm from the organic/inorganic volume ratio estimated from the AMS data."

> In text, we add: P8 L10: "... the mass accommodation coefficient, and **the radius of the inorganic core, which was estimated from a volume ratio between organic/inorganic from the AMS data** (Gaston et al., 2014; Hu et al., 2016). **The values for these parameters were constrained by the ambient aerosol measurements as described in Hu et al. (2016).**"

R1.6. P9, section 3.4. Is there a direct input from the coupled land-surface mode to MEGAN to include the effects of modeled soil moisture variabilities on monoterpene emissions or is the soil moisture emission activity factor insensitive to the range of soil moisture changes in this environment?

Since the MEGAN model is based on Guenther et al., (2006, 2012), the influence of soil moisture is only considered for isoprene and not on the emissions of other BVOCs (Sakulyanontvittaya et al., 2008 and Guenther et al., 2012). This is mentioned in Appendix B2 P39 L1/2 where soil moisture emission factor for monoterpenes is set at 1, while for isoprene is seen on P38 L18/19.

> We have clarified the text in P39 L1/2 as follows: "**The soil emission factor is, however, only considered for isoprene and not other BVOCs in the MEGAN model, and hence** for monoterpenes is set at 1 **(Sakulyanontvittaya et al., 2008; Guenther et al., 2012). "**

R1.7. P15, L6: it's mentioned that ISOPOOH SOA peak is not captured in the model, but no explanation is provided.

The disparity between the modelled and measured ISOPOOH-SOA is about 0.05 ug m$^{-3}$, and as already stated (P5 L8) "ISOPOOH-SOA is otherwise within the range of

observations." Since this is such a small difference, we prefer to not speculate about its causes.

Hence, we do not make any additions/changes to the text.

P18, L5: How is the entrainment curve in Fig. 8 determined?

We determine entrainment similar to equation A3 from Janssen et al 2012:

$$\frac{dOA_{BG}}{dt} = \frac{w_e \Delta OA_{BG}}{h}$$

The entrainment term is calculated as the change of the background OA over time. The equation represents the entrainment flux which is calculated from the entrainment velocity ($w_e$ in m s$^{-1}$), the concentration jump in background OA ($\Delta OA_{BG}$ in ug m$^{-3}$) between the RL and BL, and the boundary layer height (h in m):

> We have added the following text to clarify this point and added the entrainment equation in section 6, P18 L3
> **"The entrainment budget for the background OA is calculated as per Janssen et al., (2012):**
>
> $$\frac{dOA_{BG}}{dt} = \frac{w_e \Delta OA_{BG}}{h} \quad \textbf{(1)}$$
>
> **in which the entrainment flux is calculated from the entrainment velocity ($w_e$ in m s$^{-1}$), the concentration jump in background OA ($\Delta OA_{BG}$ in ug m$^{-3}$) between the RL and BL, and the boundary layer height (h in m)."**

R1.8. P19, L2: Monoterpenes have a continuous source as long as there is light and warm temperature, so I don't think the statement in this sentence that most of monoterpenes have reacted in the morning makes sense. Please clarify.

> Changed text to say: "**The mixing ratios of monoterpenes decrease due to entrainment in the early morning and strong reaction with OH, which peaks around noon, while the emissions, though continuous, are unable to compensate for the increased oxidation and entrainment and therefore the monoterpene contribution to SOA later in the afternoon is smaller.**"

R1.9. P19, L16: I would clarify that OH is the most important daytime oxidant during for SOA formation.

Changed text to say: "The oxidant + BVOC pathway contribution can be seen in Figure C6; **OH is the most important daytime oxidant for SOA formation.**"

R1.10. P20, L5-12: looking at the diurnal pattern of LO-OOA, it actually appears that LO-OOA concentration decreases in early morning and then increases, suggesting that

LO-OOA is locally produced. This trend is in contrast to the trend in MO-OOA, so doesn't this suggest that MO-OOA is actually more regional and present at higher conc. in the RL than the BL? This is mentioned in the conclusions, but not in this section. I think both of these trends are worth highlighting again here.

This is a good point so we added the following text to address it:

> P20 L7: "availability for G/P partitioning. **The drop in measured LO-OOA concentrations (in the morning) indicates a dilution that is driven by entrainment as LO-OOA poor air is introduced in the BL from the RL.** If we consider […]"

> P20 L10: "**The more oxidized oxygenated organic aerosol (MO-OOA) could result from entrainment from the RL though** available measurements show that the OA […] **so** not all the aged MO-OOA can be explained by this process…"

R1.11. P21, L5: Does the impact of higher air temperature on soil moisture content and response of vegetation's BVOC emissions to changes in soil moisture content also represented in the climate sensitivity runs? If not, how would such results be different?

The higher air temperature (and skin temperature) does impact the soil moisture content through evaporation (Latent heat flux (LE), especially soil component of LE) which changes with changes in temperature. As we can see below, a 2 K change in temperature does not have much of an effect on the LE, though it changes the shape of the curve, which could affect the BVOC curve. However, as mentioned in the answer to comment R1.6 (monoterpene and soil moisture), the soil moisture content effect on BVOC emissions is considered only for isoprene emissions, and this change is definitely represented in the sensitivity runs. We present the effect of changing LE has on the soil moisture, though it would be quite interesting to explore the role of soil moisture on the BVOCs and SOA formation further. This will be definitely be possible to do with the coupled land-atmosphere model we have now developed and would be useful for future studies.

[Figure]

Figure 1: The latent heat flux diurnal evolution with the case study air temperature (296.6) and 2K warmer, at SOAS campaign.

We have added the following text to address this point:

> P22 between L30/31: "**The coupled land-atmosphere model gives us the ability to explore the sensitivity of SOA formation to different variables that might change in the future due to changing climate regimes. It would be interesting, for instance to study the effect of drier or wetter climates on the SOA diurnal variability.**"

R1.13. Editorial comments: P2, L20: Another recent paper discussing importance of monoterpenes in SE-US by Xu et al. (ACP, 2018) should also be referenced. (Xu, L., Pye, H. O. T., He, J., Chen, Y., Murphy, B. N., and Ng, N. L.: Experimental and model estimates of the contributions from biogenic monoterpenes and sesquiterpenes to secondary organic aerosol in the southeastern United States, Atmos. Chem. Phys., 18, 12613-12637, https://doi.org/10.5194/acp-18-12613-2018, 2018.)

P2 L20: "… Zhang et al., 2018**; Xu et al., 2018**)."

R1.14. P5, L6: add ". . .(model results) are shown. . ."

Changed text to " … (model **results**) are shown…"

R1.15. P5, L15: change "sides" to "sites"

Thanks for catching the typo, now changed to "**sites**"

R1.16. P5, L25: remove one "been"

Omitted, "**been**"

R1.17. P7. Numbering for tables and figures seems not to be in the right order

Figure 7 appears earlier in text than Fig C4 (supplementary). Tables reordered to appear in the right order. Table B6 is now Table B3.

Added text on P6L5 "... in Vilà-Guerau de Arellano et al. (2015). **The dynamics and boundary conditions can be seen in Table B1.** In this section, …"

R1.18. P7, L26: "an early morning. . ."

P7L26 Now changed to "**an** early morning..."

R1.19. P8, L17: I would remove "the" of "the emissions. . ."

Omitted "the"

R1.20. P9, L9: consider changing "," to ";" or start a new sentence

"…interactively**;** we instead …"

R1.21. P19, L30: ". . .leads to overestimated values compared to . . ."

". . .leads to overestimate**d values** compared to . . ."

**Anonymous Referee #2**

R2.0. The authors use existing parameterisations to update a coupled land-atmosphere model to represent biogenic secondary organic aerosol (SOA) formation at a site in the southeast US. The paper is clearly written and the figures and tables are well presented. The topic is also appropriate for ACP, but the justification for the study is weak.

R2.1. The authors use mechanisms and processes that have already been applied to regional and global chemistry transport models (CTMs) (Pye et al., 2010, 2013; Marais et al., 2016) and used to describe the processes that contribute to biogenic SOA formation. It's not apparent why a land model is needed to improve the SOA simulation or that conclusions from this study couldn't be derived without the land model. There is also no evidence or

reference to past studies to support the need for the land model. What happens to the simulations when input from the land model is prescribed rather than modelled explicitly?

This is a valid point, and this allows us to clarify the reasons why we used a coupled land-atmosphere model to study the daytime SOA evolution. First, we show in this study the importance of the role of entrainment in shaping the daytime evolution of SOA in the boundary layer during SOAS. It is shown in Fig 9 that the strong SOA production during daytime is counteracted by entrainment, resulting in a flat diurnal evolution, which could not be explained by other models (e.g. Zhang et al., 2018). The representation of boundary layer dynamics and entrainment of reactants is usually a challenge for global and regional chemistry transport models. If this process is inaccurately represented it leads to errors in the entrainment fluxes for chemical species, which will lead to errors in their boundary layer concentrations. Here, we address this issue with a local-scale model which treats entrainment in an explicit way and which is well constrained by observations. The entrainment process has been shown before to be important for the correct representation of the diurnal cycle of biogenic SOA (Janssen et al., 2012, 2013), and in those studies, the fluxes of heat and VOC's were indeed prescribed.

In this work, we attempt to take this approach one step further and to make the fluxes a function of the coupled land-atmosphere system. The main reason for doing so, is the fact that the boundary layer and the land surface form a tightly coupled system (e.g. Betts, 2004, Van Heerwaarden et al, 2009), and that the fluxes respond to changes in the coupled system. So if we want to understand how VOC fluxes, heat fluxes and ultimately SOA concentrations respond to changing forcings, we need to study the response of the whole system.

The sensitivity analysis to changing temperatures in Section 8 is one example of such an analysis, since temperature can affect all surface fluxes. We can for instance consider the response of BVOC emissions, which are driven by land surface properties. To properly simulate a warmer climate, we need to include changes in temperature not only in the air, but in the canopy layer (skin temperature), and the soil layers, which are part of the land surface model. The main possibility a fully-coupled land surface model gives, however, is the possibility to look at the response of the individual variables of the whole system to a warmer climate and the contribution of these variables to SOA formation (dampening or strengthening). The conclusion that SOA concentrations are buffered against temperature changes is not something that could be foreseen otherwise.

We have added the following text to document these points:

P1, L15 insert: "**Entrainment of aged SOA from the residual layer likely contributes to the observed more oxidised oxygenated organic aerosol (MO-OOA) factor.**"

P9 L1 "**The land surface and the boundary layer form a tightly coupled system, in which fluxes respond to changes in forcings on the whole system (Betts, 2004, Van Heerwaarden et al., 2009). To properly understand the response of SOA**"

**formation to changing temperatures (see Section 8), it is therefore important to have a fully coupled land surface-boundary layer model. This allows us to study the effects on SOA evolution of a forcing which affects the coupled land-atmosphere. For that purpose,** a land surface model (Van Heerwaarden et al., 2009) is coupled to MXLCH-SOA **to obtain a fully coupled land surface-boundary layer model that** enables the interactive calculation of surface heat and entrainment fluxes, …".

P9 L5 " **Additionally, the coupled land surface model also provides input for calculating BVOC emissions interactively (see Section 3.3).**"

P23, L29 insert: "**Overall, the relatively flat diurnal cycle of the total observed SOA can be explained by the contrasting effects of local SOA production and entrainment of SOA-depleted air from the residual layer.**"

P24 L26 "**The use of a fully coupled land surface-boundary layer model that enables the interactive calculation of surface heat and entrainment fluxes, which makes it possible to study how VOC fluxes, heat fluxes and ultimately SOA concentrations respond to changing forcings.**"

R2.2. The model that the authors use also only simulates atmospheric composition for a limited time period during the day. It's not clear why this is the case and what effect this has on simulation of biogenic SOA. The output from the model also gives the impression that there is something faulty with the model, especially in Figure 5 (b) where it appears simulated isoprene mixing ratios would continue to increase beyond 3pm (when the last output is obtained from the model) and so far exceed the measurements.

We focus on a sub-diurnal scale, because the MXLCH model is valid under convective conditions only, when the boundary layer can be considered well-mixed. The model stops when the sensible heat flux has reached 0, and convection ceases. The modelled sensible heat flux is within the range of the measurements, though on the lower side of the range in the afternoon.

The majority of the daytime SOA formation has been captured in this time range as the boundary layer growth has already happened and the height of the boundary layer is rather constant. The nighttime SOA formation is not modelled. However, the model is initialised with the SOA concentration in the early morning. See also the response to comment R1.1 above for further discussion of this topic and the resulting modifications to the text.

As to isoprene, it has a large mixing ratio at the end of the day, due to high emissions. The modelled isoprene mixing ratio are actually somewhere between the mixing ratios measured in the vertical profiles and the mixing ratios measured by the tower that takes measurement above the canopy. This disparity between observations occurs since the mixing in the boundary layer is more vigorous than at the surface, which especially affects species with shorter chemical lifetimes. The model, however, assumes that the boundary layer is perfectly mixed, even at the surface. In addition, model OH concentrations are in the low range of the observations in the afternoon, which could contribute to overestimation of the isoprene

concentrations. We cannot speculate about what happens after 3pm, since the model is not valid anymore under those conditions.

> For isoprene we have modified the paper text as follows:

> P13 L9-11: "However, they are overestimated in the late afternoon **compared to the vertical profiles (green;** boosted due to the high emissions 10 calculated above the AABC tower; modelled 6.4 ppb while measurements indicate 4 ppb ±0.8 ppb), **but are better matched to the isoprene mixing ratios measured** at the SEARCH tower better (blue). The difference in measured isoprene mixing ratios indicates that isoprene is not very homogeneously well mixed in the horizontal or vertical**, while the model assumes it is. In addition, model OH concentrations are in the low range of the observations in the afternoon, which could contribute to overestimation of the isoprene concentrations.**"

R2.3. There are also inappropriate references provided to indicate the source of parameterizations and variables for the biogenic SOA formation mechanisms. For example, the authors quote Hu et al. (2016) numerous times (P2 L17, P2 L19, and throughout Section 3.2, but the variables and parameterisations are not original to that study.

One of the main reasons Hu et al., 2016 is referenced in accordance to IEPOX-SOA formation (alongside Gaston et al., 2014; which is credited for the resistor model and its parameters) is that the variables/measurements needed to model reactive uptake for the SOAS campaign are stated in this paper, as stated in response to comment R1.5.

Hence we keep the Hu et al., 2016 in P2 L17 as it is the main study that uses the resistor model and looks at IEPOX-SOA for the SOAS campaign.

However, the ISOPOOH-SOA reactions rates are from Krechmer et al., 2015, hence we make the following changes to text:

> P2 L19: omit "**Hu et al., 2016**" ..

> P7 L3: "IEPOX-SOA and ISOPOOH-SOA **(Hu et al., 2016; Krechmer et al., 2015)**

> P7 L7: "formation (upon condensation) is included using reaction rates from **Krechmer et al. (2015)**"

> P7 L21: formation through condensation and reactive uptake, which are assumed to form low-volatility aerosol products (**Krechmer et al., 2015;** Hu et al., 2016;"

> P8 L4: replace "(Hu et al., 2016)" with "**(Paulot et al., 2009)**"

> P8 L7/8: A heterogeneous reaction rate for IEPOX-SOA formation is calculated using a modified resistor model from **Gaston et al. (2014) and using inputs from Hu et al., 2016 to represent SOAS conditions**;"

**References for these responses:**

Betts, A. K. (2004). Understanding Hydrometeorology Using Global Models, Bull. Amer. Meteor. Soc., 85(11), 1673–1688, doi:10.1175/BAMS-85-11-1673.

Gaston, C. J., Riedel, T. P., Zhang, Z., Gold, A., Surratt, J. D., and Thornton, J. A. (2014). Reactive Uptake of an Isoprene-Derived Epoxydiol to Submicron Aerosol Particles, Environ. Sci. Technol., 48, 11178–11186, doi:10.1021/es5034266.

Guenther, A. B., Jiang, X., Heald, C. L., Sakulyanontvittaya, T., Duhl, T., Emmons, L. K., & Wang, X. (2012). The Model of Emissions of Gases and Aerosols from Nature version 2.1 (MEGAN2. 1): an extended and updated framework for modeling biogenic emissions.

Hu, W. W., Campuzano-Jost, P., Palm, B. B., Day, D. A., Ortega, A. M., Hayes, P. L., Krechmer, J. E., Chen, Q., Kuwata, M., Liu, Y. J., De Sá, S. S., McKinney, K., Martin, S. T., Hu, M., Budisulistiorini, S. H., Riva, M., Surratt, J. D., St. Clair, J. M., Isaacman-Van Wertz, G., Yee, L. D., Goldstein, A. H., Carbone, S., Brito, J., Artaxo, P., De Gouw, J. A., Koss, A., Wisthaler, A., Mikoviny, T., Karl, T., Kaser, L., Jud, W., Hansel, A., Docherty, K. S., Alexander, M. L., Robinson, N. H., Coe, H., Allan, J. D., Canagaratna, M. R., Paulot, F., and Jimenez, J. L. (2015). Characterization of a real-time tracer for isoprene epoxydiols-derived secondary organic aerosol (IEPOX-SOA) from aerosol mass spectrometer measurements, Atmospheric Chemistry and Physics, 15, 11 807–11 833, https://doi.org/10.5194/acp-15-11807-2015

Hu, W., Palm, B. B., Day, D. A., Campuzano-Jost, P., Krechmer, J. E., Peng, Z., ... & Hacker, L. (2016). Volatility and lifetime against OH heterogeneous reaction of ambient isoprene-epoxydiols-derived secondary organic aerosol (IEPOX-SOA). *Atmospheric Chemistry and Physics*, *16*(18), 11563-11580.

Janssen, R., Vilà-Guerau de Arellano, J., Ganzeveld, L., Kabat, P., Jimenez, J., Farmer, D., Van Heerwaarden, C., and Mammarella, I. (2012) Combined effects of surface conditions, boundary layer dynamics and chemistry on diurnal SOA evolution, Atmospheric Chemistry and Physics, 12, 6827–6843.

Krechmer, J. E., Coggon, M. M., Massoli, P., Nguyen, T. B., Crounse, J. D., Hu, W., Day, D. A., Tyndall, G. S., Henze, D. K., Rivera-Rios, J. C., Nowak, J. B., Kimmel, J. R., Mauldin, R. L., Stark, H., Jayne, J. T., Sipilä, M., Junninen, H., St. Clair, J. M., Zhang, X., Feiner, P. A., Zhang, L., Miller, D. O., Brune, W. H., Keutsch, F. N., Wennberg, P. O., Seinfeld, J. H., Worsnop, D. R., Jimenez, J. L. and Canagaratna, M. R. (2015). Formation of Low Volatility Organic Compounds and Secondary Organic Aerosol from Isoprene Hydroxyhydroperoxide Low-NO Oxidation, Environ. Sci. Technol., 49(17), 10330–10339, doi:10.1021/acs.est.5b02031.

Marais, E. A., Jacob, D. J., Jimenez, J. L., Campuzano-Jost, P., Day, D. A., Hu, W., Krechmer, J., Zhu, L., Kim, P. S., Miller, C. C., Fisher, J. A., Travis, K., Yu, K., Hanisco, T. F., Wolfe, G. M., Arkinson, H. L., Pye, H. O. T., Froyd, K. D., Liao, J., and Mcneill, V. F.

(2016). Aqueous-phase mechanism for secondary organic aerosol formation from isoprene: application to the southeast United States and co-benefit of SO2 emission controls, Atmos. Chem. Phys, 16, 1603–1618, https://doi.org/10.5194/acp-16-1603-2016, www.atmos-chem-phys.net/16/1603/2016/.

Paulot, F., Crounse, J. D., Kjaergaard, H. G., Kürten, A., Clair, J. M. S., Seinfeld, J. H., & Wennberg, P. O. (2009). Unexpected epoxide formation in the gas-phase photooxidation of isoprene. *Science*, *325*(5941), 730-733.

Sakulyanontvittaya, T., Duhl, T., Wiedinmyer, C., Helmig, D., Matsunaga, S., Potosnak, M., ... & Guenther, A. (2008). Monoterpene and sesquiterpene emission estimates for the United States. *Environmental science & technology*, *42*(5), 1623-1629.

Van Heerwaarden, C. C., Vilà-Guerau de Arellano, J., Moene, A. F. and Holtslag, A. A. M. (2009). Interactions between dry-air entrainment, surface evaporation and convective boundary-layer development, Quart. J. Roy. Meteor. Soc., 135(642), 1277–1291, doi:10.1002/qj.431.

Xu, L., Pye, H. O. T., He, J., Chen, Y., Murphy, B. N., and Ng, N. L. (2018). Experimental and model estimates of the contributions from biogenic monoterpenes and sesquiterpenes to secondary organic aerosol in the southeastern United States, Atmos. Chem. Phys., 18, 12613-12637, https://doi.org/10.5194/acp-18-12613-2018.

Yee, L. D., Isaacman-VanWertz, G., Wernis, R. A., Meng, M., Rivera, V., Kreisberg, N. M., ... & Gray Bé, A. (2018). Observations of sesquiterpenes and their oxidation products in central Amazonia during the wet and dry seasons. *Atmospheric Chemistry and Physics*, *18*(14), 10433-10457.

Zhang, H., Yee, L. D., Lee, B. H., Curtis, M. P., Worton, D. R., Isaacman-VanWertz, G., Offenberg, J. H., Lewandowski, M., Kleindienst, T. E., Beaver, M. R., Holder, A. L., Lonneman, W. A., Docherty, K. S., Jaoui, M., Pye, H. O. T., Hu, W., Day, D. A., Campuzano-Jost, P., Jimenez, J. L., Guo, H., Weber, R. J., Gouw, J. de, Koss, A. R., Edgerton, E. S., Brune, W., Mohr, C., Lopez-Hilfiker, F. D., Lutz, A., Kreisberg, N. M., Spielman, S. R., Hering, S. V., Wilson, K. R., Thornton, J. A. and Goldstein, A. H. (2018). Monoterpenes are the largest source of summertime organic aerosol in the southeastern United States, PNAS, 201717513, doi:10.1073/pnas.1717513115.